# A Multi-Regional Path-Planning Method for Rescue UAVs with Priority Constraints

Lexu Du [1,2], Yankai Fan [3], Mingzhen Gui [1,2] and Dangjun Zhao [1,2,*]

1   School of Automation, Central South University, Changsha 410083, China; dulx3364@csu.edu.cn (L.D.);
    guimingzhen@csu.edu.cn (M.G.)
2   Hunan Provincial Key Laboratory of Optic-Electronic Intelligent Measurement and Control,
    Changsha 410083, China
3   Academy of Astronautics, Nanjing University of Aeronautics and Astronautics,
    Nanjing 211106, China; yankai.fan@nuaa.edu.cn
*   Correspondence: zhao_dj@csu.edu.cn

**Abstract:** This study focuses on the path-planning problem of rescue UAVs with regional detection priority. Initially, we propose a mixed-integer programming model that integrates coverage path planning (CPP) and the hierarchical traveling salesman problem (HTSP) to address multi-regional path planning under priority constraints. For intra-regional path planning, we present an enhanced method for acquiring reciprocating flight paths to ensure complete coverage of convex polygonal regions with shorter flight paths when a UAV is equipped with sensors featuring circular sampling ranges. An additional comparison was made for spiral flight paths, and second-order Bezier curves were employed to optimize both sets of paths. This optimization not only reduced the path length but also enhanced the ability to counteract inherent drone jitter. Additionally, we propose a variable neighborhood descent algorithm based on K-nearest neighbors to solve the inter-regional access order path-planning problem with priority. We establish parameters for measuring distance and evaluating the priority order of UAV flight paths. Simulation and experiment results demonstrate that the proposed algorithm can effectively assist UAVs in performing path-planning tasks with priority constraints, enabling faster information collection in important areas and facilitating quick exploration of three-dimensional characteristics in unknown disaster areas by rescue workers. This algorithm significantly enhances the safety of rescue workers and optimizes crucial rescue times in key areas.

**Keywords:** UAV coverage path planning; traveling salesmen problem; priority constraints; path optimization

## 1. Introduction

Unmanned aerial vehicles (UAVs) have been widely used in many domains due to their small size, sensitive control, and high scalability. Especially in the civil field, UAVs equipped with cameras, infrared, LiDAR, or other sensors can conduct various missions, including personnel searches [1], field monitoring [2], and terrain detection [3]. As UAVs can survey target areas without causing any damage, they are well-suited to performing information sensing tasks in remote or hazardous environments and complex terrains. By reaching a rescue site before rescue personnel, drones provide timely and precise target information, greatly enhancing the efficiency of rescue operations. To this end, a reasonable path should be planned such that the mobile sensor carried by a UAV can cover a region in a finite time, giving rise to a coverage path-planning (CPP) problem with various constraints, including energy consumption, time consumption, etc.

In recent years, extensive research on the CPP problem for covering single regions with energy constraints and photography constraints has been carried out [4–8]. A covered route primarily exhibits various shapes, including round-trip and spiral patterns. However,

the UAV's motion capabilities in actual flight are limited, making it challenging to precisely follow sharp corners in simulated routes at turning points. Therefore, smoothing out the sharp corners of the flight path becomes essential to save UAV flight time and reduce jitter during turning. This challenge is extensively studied in the realm of UAV obstacle avoidance in flight and is also applicable in UAV path-planning route optimization. The Bezier curve was initially widely employed in robot motion planning. In recent years, its application has expanded to include the field of UAVs. Machmudah et al. [9] studied the incline and turn flight trajectory optimization of fixed-wing UAVs at a fixed altitude. Utilizing the Bezier curve as the maneuvering path, the speed change reduces the load coefficient of the inclined steering mechanism, and a simultaneous on-arrival target mission has also been successfully conducted when the turning radius was small. However, in practical scenarios of large-scale search and rescue missions, it is often not feasible to consider the entire disaster area as a single region for coverage. Instead, the area is divided into multiple areas of interest (AOIs) based on disaster information. Multiple disaster locations are then selectively surveyed to effectively obtain post-disaster information. Therefore, the challenge of path planning to cover multiple regions becomes a compelling research topic.

In general, the multi-regional path-planning problem can be converted to a combination of two subproblems: a traveling salesmen problem (TSP) and a CPP problem [10], constituting a TSP-CPP problem. A two-step path-planning method [11] is proposed to cover multiple disjoint regions: (1) the access order of UAVs between regions is determined in the first step by using genetic algorithms, and (2) the coverage path inside a region is determined in the second step by using the rotating caliper algorithm [12]. Xie et al. [10] planned the coverage path for multiple two-dimensional rectangular areas based on the grid approach and dynamic programming methods. The target area is first split into mesh grids according to the sensor's sampling range; thus, the original TSP-CPP problem is converted to a TSP problem, which is then solved by a dynamic programming method. Further, the authors proposed a heuristic algorithm based on NN-2Opt [13] for efficiently covering all regions even when there is a large number of regions. In [14], the minimum distance strategy was considered, and an improved simulated annealing algorithm was proposed to determine the access sequence of multiple regions, after which a back-and-forth (BF) path is generated to cover multiple convex polygonal regions. Ko et al. [15] proposed a novel UAV trajectory-planning method to optimize location-dependent visual coverage. In this method, the UAV dynamically adjusts its altitude to meet varying image-resolution requirements. Comprising three components, the approach effectively minimizes task completion time.

Indeed, in numerous disaster relief missions, the significance of regions is determined by factors such as severity, distance from the disaster center, and population density. While the aforementioned methods have effectively tackled the traditional TSP-CPP problem, they have failed to consider the diverse priorities of multi-target regions inherent in many large-scale rescue missions. To date, there has been scarce research on path-planning problems that accommodate distinct priorities for different regions. Miao et al. [16] introduced UAV-assisted moving edge computing (MEC) using UAVs as MEC nodes and proposed a multi-UAV-assisted MEC unloading algorithm based on global and local path planning. The approach takes into account the priority of monitoring sites but focuses on optimizing drone swarm scheduling, distribution, and communication coverage to minimize flight length and energy consumption. In [17], the access order of multiple regions is manually prescribed, and a heuristic algorithm is used to generate the sequence of regions without considering the different priority levels of different regions.

In order to deal with priority constraints, the planned access order for multiple regions depends on the prescribed priority levels of each region; thus, the original TSP should be extended as a hierarchical traveling salesman problem (HTSP) [18]. In [19], regions with the same priority are clustered into one cluster; correspondingly, regions with different priorities are clustered into different clusters, whose access sequences are determined

by their priority levels. However, this planning approach ensures that the accesses are planned in order of priority, but it invariably results in a significant degree of path-length redundancy. In some application scenarios, this strategy may not be the most efficient and effective. Panchamgam et al. [18] proposed a d-relaxed priority model, in which priority was adjusted to a certain extent during the planning process while taking path length into account. The rule of this model is as follows: if $p$ is the highest priority of all unvisited locations, the vehicle is allowed to access one of the positions whose priority is $p$, $p + 1, \ldots, p + d$. The value of the positive integer $d$ can be flexibly controlled to realize the trade-off between path cost and emergency degree. Hà et al. [20] established a d-relaxed priority integer programming model based on [18] and introduced a metaheuristic method based on the framework of iterated local search with problem-tailored operators to find approximate solutions.

In the realm of drone path planning for real-life rescue scenarios, a comprehensive and systematic approach for multi-regional path-planning tasks with priority constraints is lacking. Such a method should have the ability to determine priority sequences and path lengths tailored for evaluating emergency rescue tasks effectively. This paper presents the following contributions: building upon the work presented in [18], we formally define the HTSP-CPP problem and formulate it as a mixed-integer programming model with d-relaxed constraints. In the realm of intra-regional path planning, we present an enhanced BF path coverage method that leverages the minimum width of polygonal regions. This method ensures complete coverage of convex regions by sensors equipped with circular sampling ranges. And simulate a comparison with the spiral path, optimizing both paths using Bezier curves. To optimize inter-regional access order planning, we introduce two different strategies for generating initial solutions, enabling efficient determination of the access sequence for multiple regions, and utilizing the RVND algorithm to optimize the initial solution. Additionally, we propose a distance–priority evaluation rule to assess the trade-off between distance and priority with respect to the solutions.

The structure of this paper is as follows. In Section 2, we present a mixed-integer programming model based on the HTSP-CPP. In Section 3, we discuss the related algorithms and explain the specific algorithm design and process for intra-regional and inter-regional path planning. We also propose a result evaluation index during the experimental design stage, design a specific simulation scheme, and present the simulation and experiment results in Sections 4 and 5.

## 2. Mathematical Model of the HTSP-CPP

The sensors integrated into UAVs mainly consist of LiDARs, RGB cameras, NIR cameras, and others. Studies by Salach et al. [21] and Domingo et al. [22] have highlighted LiDARs' superiority in terrain detection and 3D modeling compared to other sensors. However, concerning search and rescue missions, NIR cameras and RGB cameras demonstrate more significant potential. In complete disaster relief operations, UAVs equipped with LiDAR are initially utilized to gather terrain data in affected regions, aiding in disaster severity assessment and the formulation of relief strategies. Subsequently, UAVs outfitted with infrared or optical sensors conduct a secondary search in areas where individuals may be trapped, precisely identifying their locations for efficient rescue efforts. Therefore, the path-planning method should be applicable to various sensor types. However, due to the unique sampling shape of LiDAR, this paper focuses on improving the coverage method based on circular sampling ranges. Table 1 shows the main symbols used in this article.

**Table 1.** Summary table of important symbols.

| Symbol | Definition | Symbol | Definition |
|--------|-----------|--------|-----------|
| $A_i$ | The region's parameter set | $d$ | The coefficient of relaxation |
| $C$ | The set of accessed regions | $d_L$ | The distances from the strip's sides to the base edge |
| $C_i$ | The central point coordinates of the region | $e_p^i$ | Decision variable, whether the waypoint, $p$, is the entrance of the region |
| $\mathbf{D}$ | The distance matrix | $g$ | The number of priorities |
| $E$ | The set of unvisited regions | $m_i$ | The number of waypoints in the region |
| $G$ | The priority set | $n_i$ | The number of vertices in the region |
| $L_i$ | The sides of the strip proximal to the bottom | $o$ | The path overlap rate of the UAV |
| $L_i'$ | The sides of the strip further to the bottom | $p_i$ | The priority of the region |
| $N$ | The region's number set | $t_p^i$ | Decision variable, whether the waypoint, $p$, is the export of the region |
| $N_0$ | The region's number set including the depot | $u_i$ | The position of the region in the access sequence |
| $N_b$ | The number of stripes in the region | $v_{ij}$ | The vertex coordinates of the region |
| $\mathbf{P}$ | The priority matrix | $w_{ij}$ | The coordinate of the waypoints |
| $R_k$ | The set of regions with the same priority | $x_{ij}$ | Decision variable, whether there is a connecting path between two regions, $i, j$ |
| $S$ | The width of the convex region | $y_{pq}^i$ | Decision variable, whether there is a connecting path between two waypoints, $p, q$, in the region |
| $V_i$ | The vertex set of the region | $\omega$ | The sensor's sampling diameter |
| $W_i$ | The waypoints set of the region | $\Delta d$ | The distance between the UAV's scan lines |

### 2.1. Problem Description

We assume that there are $n$ convex polygonal regions with different sizes and shapes to be covered, and these regions are denoted by $A_i = <V_i, C_i, p_i>$, where $V_i = \{v_{i1}, v_{i2}, \ldots, v_{in_i}\}$ is the vertex coordinates of the clockwise arrangement of polygons, $v_{ij}$ is the coordinates of the $j$-th vertex of the $i$-th region, $n_i$ is the number of vertices in the $i$-th region, $C_i$ is the central point coordinate of the $i$-th region, and $p_i$ is the priority of the $i$-th region, where $p_i \in G = \{1, \ldots, g\}$.

Now, a UAV equipped with a sensor (LiDAR) is utilized to initiate a comprehensive coverage detection of these $n$ areas starting from the base for acquiring the height and obstacle information of each region. Upon completion of the coverage task, the UAV returns to the base. The ground sampling range of the sensor is circular in shape, as depicted in Figure 1.

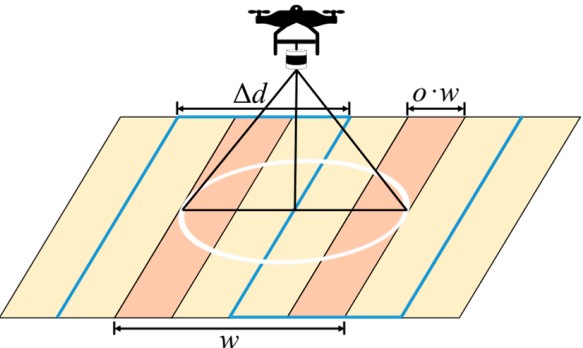

**Figure 1.** Ground sampling range of sensor.

The sampling diameter of the sensor is influenced not only by the sensor's performance, including the field of view angle, the angular resolution, and the maximum detection radius, but also the flight parameters, such as the flight height and velocity. As the flight altitude increases, the sampling range expands while the range that satisfies the required sampling precision decreases. Therefore, there exists a maximum flight altitude that ensures adequate sampling precision. Similarly, there exists a minimum flight altitude that guarantees the minimum required sampling range. When the UAV operates within this permissible altitude range, minor fluctuations in altitude will not impact sampling effectiveness. This allowable range is determined by both task-specific sampling precision requirements and sensor parameters. To simplify matters, we treat the sampling range necessary for achieving the desired precision as a fixed value and introduce a concept of "sampling overlap rate" to ensure that UAV altitude changes during flight do not compromise the task's overall quality of data collection. The sampling diameter is $w$, and the distance between scan lines is given as:

$$\Delta d = \omega(1 - o) \tag{1}$$

Since UAVs maintain a constant flight height regardless of the terrain's ups and downs, this problem can be considered a two-dimensional HTTP-CPP problem. Assuming that the sensor's sampling footprint can completely cover the target region, the aggregate flight path length of the UAV is taken as the flight cost, and the access sequence between regions and the flight path within each region are the decision variables to be optimized for minimizing the flight cost.

*2.2. Problem Modelling*

The path of a UAV is generated by a set of a series of waypoints $W_i = \{w_{i1}, w_{i2}, \cdots, w_{im_i}\}$, $i \in \{1, 2, \cdots, n\}$, where $n$ is the number of target areas and $m_i$ represents the number of waypoints in the $i$-th region, indicating the number of elements in the point set $W_i$. Each waypoint signifies a change in direction for the UAV. It is important to note that the UAV maintains a straight-line trajectory between any two consecutive waypoints. We divide the flight paths into two types: (1) intra-regional paths in a single region and (2) inter-regional paths connecting different regions.

a.    Intra-regional paths

Let $y^i_{pq}$ denote the access order of waypoints $p, q \in \{1, 2, \cdots, m_i\}$ of the intra-regional path for the $i$-th region and $y^i_{pq} = 1$ indicate that the UAV flies from point $p$ to point $q$, while $y^i_{pq} = 0$ indicates that there is no connecting path between the waypoints $p$ and $q$. Then, let $e^i_p$ and $t^i_p$ denote the import and export of the $i$-th region, respectively. When $e^i_p = 1$ means that the UAV flies into region $i$ from the waypoint $p$, $e^i_p = 0$ means that the waypoint $p$ is not the entry point in region $i$. Similarly, when $t^i_p = 1$ represents that the UAV flies out of region $i$ from the waypoint $p$, $t^i_p = 0$ represents that the waypoints $p$ is not the exit point in region $i$.

b.    Inter-regional paths

The priority constraints should be imposed on the inter-regional path planning to obtain the optimal access sequence of multiple regions. Intending to cover high-priority regions as extensively as possible while minimizing flight costs, it becomes necessary to slightly relax the priority of individual regions. In this paper, we employ d-relaxed priority to model this problem. The d-relaxed priority approach ensures that during the planning of an inter-regional access sequence, if the current region has a priority of $k \in G = \{1, \cdots, g\}$, the priority of the subsequent region to be accessed should not exceed $k + d$. When $d = 0$, regional access should strictly adhere to the order of priority from high to low. Conversely, when $d = g - 1$, the problem is degraded into an ordinary TSP problem without any

priority constraints. By selecting an appropriate value for *d*, a suitable compromise can be achieved between the distance traveled and the priority of regions.

Suppose the target regions are classified into different groups, $R_k$, according to their priorities, which represent the set of regions with priority level *k*. When planning the sequence of inter-regional access, an access sequence *Order* is generated. The value of $u_i$ represents the sequence number in which the *i*-th region is accessed, that is, the index value of region *i* in *Order*. Let $N = \{1, 2, \cdots, n\}$ be the set of all area numbers to be accessed and $N_0 = \{0, 1, 2, \cdots, n\}$ be the set of numbers containing the depot, where 0 is the number of the depot. A decision variable $x_{ij}$ ($i, j \in N_0$) is introduced to represent the inter-regional access order, and $x_{ij} = 1$ if the UAV flies from region *i* to region *j*, while $x_{ij} = 0$ if there is no connection path between region *i* and region *j*. Therefore, the constraints on $u_i$ and $x_{ij}$ can be expressed as follows.

$$u_i + 1 - M(1 - x_{ij}) \leq u_j, \ \forall i, j \in N_0, j \neq 0 \tag{2}$$

$$u_i + 1 < u_j, \ \forall i \in R_k, j \in R_l; \ k, l \in G, l > k + d \tag{3}$$

The first constraint states that for any two regions *i* and *j*, when region *j* is not the depot and there exists a path from *i* to *j*, then the access order of region *j* should be after region *i*. Alternatively, if there is no path from *i* to *j*, this inequality is also satisfied when a sufficiently large *M* is used. The second constraint states that when the priority of region *j* does not meet the d-relaxed constraint, the access order of region *j* should be after region *i* and not in the immediate subsequent access position after region *i*.

c. Integer programming model of the HTSP-CPP Consider the following specific scenarios:

1. In the case where the sensor's sampling range can only cover region *i*, i.e., $m_i = 1$, meaning there is only one waypoint, $w_{i1}$, in region *i*, which coincides with the center of mass of region *i*. The UAV enters and exits region *i* from $w_{i1}$ simultaneously. In other cases where $m_i > 0$, it is necessary to ensure that the UAV enters and exits region *i* from different points to avoid redundant path lengths.

2. When there are multiple regions and each region has only one waypoint, i.e., $N > 1$, then the problem simplifies to the TSP.

3. When there is only one region and its area exceeds the sampling range of the sensor, i.e., $N = 1$ and $m_1 > 0$, the problem becomes a CPP problem with a starting point and an ending point.

Taking into account the above scenarios, the objective function for this problem is as follows.

$$
\begin{aligned}
J = &\sum_{i=1}^{n} \sum_{j=1, j \neq i}^{n} \sum_{p=1}^{m_i} \sum_{q=1}^{m_j} x_{ij} t_p^i e_q^j d(w_{ip}, w_{jq}) + \sum_{i=1}^{n} \sum_{p=1}^{m_i} \sum_{q=1, q \neq p}^{m_i} y_{pq}^i d(w_{ip}, w_{iq}) \\
&+ \sum_{i=1}^{n} \sum_{p=1}^{m_i} x_{0i} e_p^i d(c_0, w_{ip}) + \sum_{i=1}^{n} \sum_{p=1}^{m_i} x_{0i} t_p^i d(w_{ip}, c_0)
\end{aligned}
\tag{4}
$$

where $d(a, b)$ represents the Euclidean distance from *a* to *b* and $c_0$ is the coordinates of the depot. In Equation (4), the first term is the path length between regions, the second term represents the sum of path lengths within all regions, the third term represents the path length from the base to the entrance of the first region, and the fourth term represents the path length from the exit of the last region back to the base.

In addition, the following constraints should be imposed on Equation (4):

$$\sum_{j=0, j \neq i}^{n} x_{ij} = 1, \ \forall i \in N_0 \tag{5}$$

$$\sum_{i=0, i \neq j}^{n} x_{ij} = 1, \ \forall j \in N_0 \tag{6}$$

$$u_i + 1 - M(1 - x_{ij}) \le u_j, \ \forall i, j \in N_0, \ j \neq 0 \tag{7}$$

$$u_i + 1 < u_j, \ \forall i \in R_k, \ j \in R_l; \ k, l \in G, \ l > k + d \tag{8}$$

$$\sum_{i \in R_k} \sum_{j \in R_l} x_{ji} = 0, \forall k, l \in G, l > k + d \tag{9}$$

$$\sum_{i \in R_k} x_{0i} = 0, \ \forall k \in G, \ k > 1 + d \tag{10}$$

$$\sum_{i \in R_k} x_{i0} = 0, \ \forall k \in G, \ k < g - d \tag{11}$$

$$\sum_{q=1, q \neq p}^{m_i} y_{pq}^i = 1 - t_p^i, \ \forall i \in N, \ p \in \{1, 2, \cdots, m_i\} \tag{12}$$

$$\sum_{p=1, p \neq q}^{m_i} y_{pq}^i = 1 - e_p^i, \ \forall i \in N, \ q \in \{1, 2, \cdots, m_i\} \tag{13}$$

$$\sum_{p=1}^{m_i} e_p^i = 1, \ \sum_{p=1}^{m_i} t_p^i = 1, \ \forall i \in N \tag{14}$$

$$\sum_{i,j \in M_1} x_{ij} \le |M_1| - 1, \ \forall M_1 \subset \{0, 1, 2, \cdots, N\}, \ 2 \le |M_1| \le N - 1 \tag{15}$$

$$\sum_{p,q \in M_2} y_{pq}^i \le |M_2| - 1, \ \forall i \in \{1, 2, \cdots, N\}, \ M_2 \subset \{1, 2, \cdots, m_i\}, \ 2 \le |M_2| \le m_i - 2 \tag{16}$$

$$x_{ij} \in \{0, 1\}, \ \forall i, j \in N_0 \tag{17}$$

$$u_i, u_j \in N_0, \ \forall i, j \in N_0, \ j \neq 0 \tag{18}$$

$$y_{pq}^i, e_p^i, t_p^i \in \{0, 1\}, \ \forall i \in N, \ p, q \in \{1, 2, \cdots, m_i\} \tag{19}$$

$$e_p^i + t_p^i \le 1, \ \forall i \in N, \ m_i > 1, \ p \in \{1, 2, \cdots, m_i\} \tag{20}$$

Equations (5) and (6) specify that the UAV enters and exits each region, including the depot, only once. This implies that each region can be visited only once. Equation (7) represents the relationship between the position variable, $u_i$, and the decision variable, $x_{ij}$. Equations (8)–(11) describe the constraint conditions of d-relaxed priority: Equation (8) defines the relationship between regional positions under the d-relaxed priority constraint and Equation (9) specifies that, when the d-relaxed priority constraint is not satisfied, a region with lower priority cannot be directly transferred to a region with higher priority. Equations (10) and (11) are the constraints of the d-relaxed priority rule when leaving the depot and returning to the depot, respectively. Equations (12) and (13) indicate that, apart from the entrance and exit of each region, for each other waypoint, the UAV will fly from one waypoint to another, ensuring that each waypoint can be accessed only once and avoiding the redundant path length caused by repeated access to waypoints. Equation (14) states that each region has one and only one entrance and exit. Equations (15) and (16) ensure the continuity of paths to eliminate subloops within and between regions, and $|M|$ represents the cardinality of set $M$. Equations (17)–(20) specify the value ranges of the decision variables to ensure their effectiveness.

## 3. Algorithm Design

Based on the aforementioned analysis, the resolution of the HTSP-CPP problem articulated by Equations (4)–(19) can be solved through the proposed algorithm, which encompasses the following three key steps:

(1)  Calculation of the inter-regional path and parameters: The initial step involves determining the optimal flight direction for the UAV within each convex polygonal region. This is achieved by calculating the width of the region. Simultaneously, the distance between flight paths is determined based on predefined sampling requirements. This information facilitates the generation of parallel flight lines within the region, subsequently yielding four potential candidate flight entry points;

(2)  Construction of the priority-constrained TSP: The algorithm designates both the center of each region and the depot as "cities" to be visited. This lays the foundation for formulating a traveling salesman problem with priority constraints. To compute the most efficient order of access, a heuristic algorithm is employed. This step aids in identifying the optimal sequence for visiting the designated cities;

(3)  Selection of optimal entry points and path generation: Utilizing the determined optimal access order, the algorithm proceeds to select the best entry points within each region. By amalgamating all selected waypoints, a coherent and comprehensive UAV path is formulated, which represents the final output of the algorithm.

In the following subsections, we will delve into the principles and code logic of this algorithm, conducting a comprehensive analysis of its effectiveness in solving HTSP-CPP problems. Additionally, we introduce a region order optimization algorithm and showcase the optimization impact on region access order using distance and priority evaluation criteria.

### 3.1. Calculation of Inter-Regional Paths and Parameters

In this paper, we employ a BF path pattern for comprehensive coverage of the designated region. Notably, executing turns with a UAV entails heightened energy consumption and concomitant augmentation of the overall path length. Consequently, diminishing the frequency of turns stands as a pivotal means of curtailing drone flight costs. Guided by the imperative of turn reduction, this paper embarks on a quest to ascertain the optimal width of convex polygons [23]. Subsequently, the UAV's BF path aligns with the vertical direction of the width of these polygons, a strategic alignment that serves to minimize path length across the targeted expanse.

a.  Calculate the width of the convex polygon to determine the best flight direction of the UAV

The authors of [23] proposed a method for determining the width of a convex polygon, which is defined as the minimum span of the polygon. This characteristic is evident specifically in polygon configurations featuring vertex–edge patterns. Notably, the span between two parallel edges can also represent the width of the polygon, which is recognized as a distinct instance of the vertex–edge scenario.

To calculate the width of a convex polygon, a comprehensive procedure is followed for each of its sides. This involves the calculation of the distances from all vertices to the selected edge. The longest calculated distance corresponds to the height of the chosen edge, thereby identifying the vertex associated with this particular height. By performing this calculation for all sides of the convex polygon and subsequently comparing the computed lengths, the minimum height emerges as the polygon's width, denoted as $S$. Remarkably, the edge linked with this minimum height establishes the optimal flight direction for the UAV [12].

The optimal flight direction is shown in Figure 2. Herein, the red edge and the green vertex are indicative of the edge and vertex corresponding to the width of the polygon, respectively. Consequently, the optimal flight orientation for the UAV is one that parallels the identified red edge.

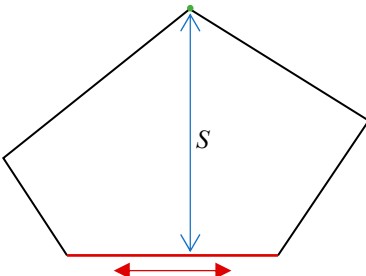

**Figure 2.** Optimum flight direction.

b. Dividing the region into strips along the width direction according to the distance between flight lines

Due to the circular sampling range of the sensor system, the scenarios depicted in Figure 3a,b are likely to arise when employing the conventional convex polygon coverage method [12]. In Figure 3, the red-shaded regions within the red boxes illustrates the uncovered area. The blue arrows denote the flight path, the green dots indicate the starting point of the flight path, and the orange area represents the coverage achieved by the sensor. To address this limitation and ensure comprehensive coverage of both vertices or edges, this paper employs a strategy involving the division of a polygon into multiple strip-like subregions along the width direction [24]. The resultant coverage effect is illustrated in Figure 3c. It becomes evident that this approach facilitates the complete coverage of all corners within the designated area, while concurrently adhering to the requisite standards for sampling quality.

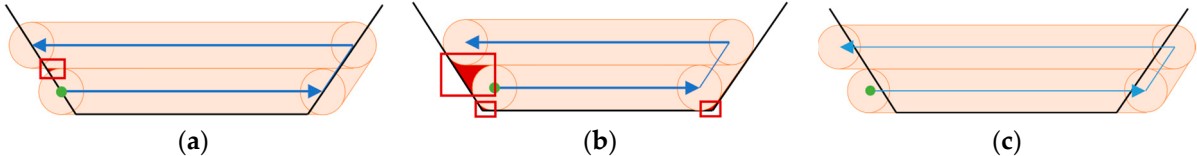

**Figure 3.** Effects of different coverage methods. (**a**) The flight line starts on the edge of the polygon. (**b**) The edge of the sensor is tangent to the edge of the polygon. (**c**) The method proposed in this paper.

The width of a strip corresponds to the scanning diameter, $w$, of the sensor. As shown in Figure 4, the median line that runs parallel to the optimal flight direction in the strip serves as the designated UAV flight trajectory. Furthermore, the intersection points between this median line and the strip delineate the specific path points for the UAV. Notably, the length of the strip plays a pivotal role in dictating the extent of the flight trajectory, thereby ensuring that the sensor can effectively encompass regions at the vertex or edge of the polygon. This strategic arrangement safeguards against scenarios akin to those depicted in Figure 3a,b.

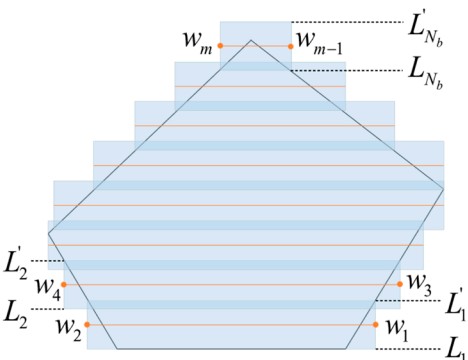

**Figure 4.** Strip division method.

For the $i$-th strip, which aligns with the best flight direction, the two parallel sides are denoted as $L_i$ and $L'_i$, respectively. Among these, $L_i$ corresponds to the side proximal to the bottom, while $L'_i$ pertains to the side situated further from the bottom. The determination of the strip's position is achieved by calculating the distances, $d_{L_i}$ and $d_{L'_i}$, from the strip's sides to the base edge. This calculation is performed according to the following equation:

$$\begin{cases} d_{L_i} = \Delta d \cdot (i - 1) \\ d_{L'_i} = d_{L_i} + \omega \end{cases} \tag{21}$$

where $\Delta d = \omega \cdot (1 - o)$, $i \in \{1, 2, \cdots, N_b\}$. $N_b$ denotes the overall quantity of stripes, and its calculation is delineated as follows:

$$N_b = \begin{cases} \left\lfloor \frac{S}{\Delta d} \right\rfloor, \ if \ S\backslash\Delta d \le \omega \cdot o \\ \left\lceil \frac{S}{\Delta d} \right\rceil, \ if \ S\backslash\Delta d > \omega \cdot o \end{cases} \tag{22}$$

The extent of a strip's length is determined by the minimal measure required for each strip to precisely encompass the polygon. This entails considering three distinct scenarios:

(1) When both edges of the strip intersect the polygon, generating two intersections, as illustrated in Figure 5a, the strip's length is essentially the greater of the distances between the two intersections. Mathematically, the length of the strip is represented by $d(I_3, I_4)$;

(2) In instances where the strip is defined by a single edge that intersects with the polygon at two points, depicted in Figure 5b, the strip's length equates to the distance between these two intersection points. This is succinctly expressed as $d(I_1, I_2)$;

(3) If a strip encompasses vertices within its scope, as illustrated in Figure 5c, the strip's length is the shortest distance capable of covering the given vertex. This length is symbolized by $d(I_5, I_6)$.

c. Generate candidate paths based on the four entry points

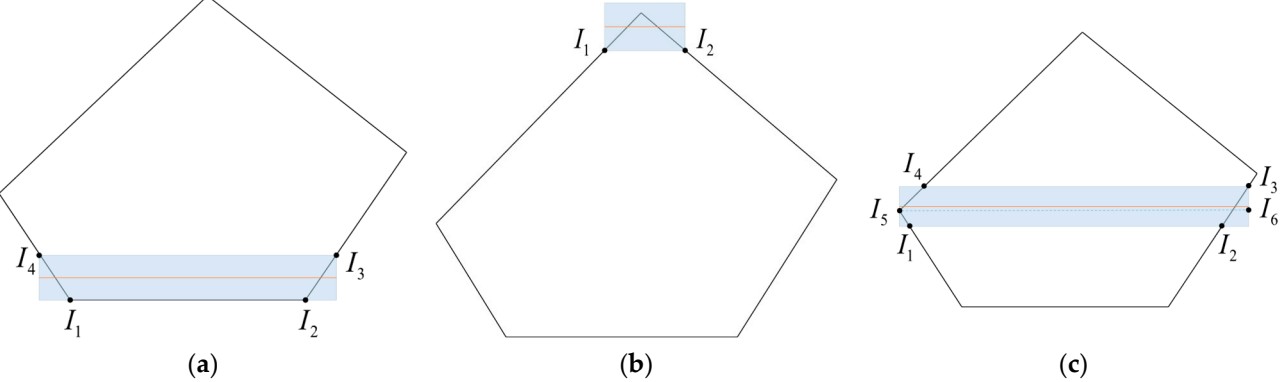

**Figure 5.** Strip length analysis. (**a**) Two edges, two points. (**b**) One edge, two points. (**c**) Strip containing vertices.

Once $N_b$ parallel scan lines have been determined, the initial and terminal points of the first scan line and the $N_b$-th scan line inherently constitute the four potential flight entry points for the current designated region. Subsequently, the UAV embarks on its coverage mission by entering the region through these four flight entry points. This sequential process engenders the creation of four distinct, complete paths confined within the region's bounds, as visually depicted in Figure 6.

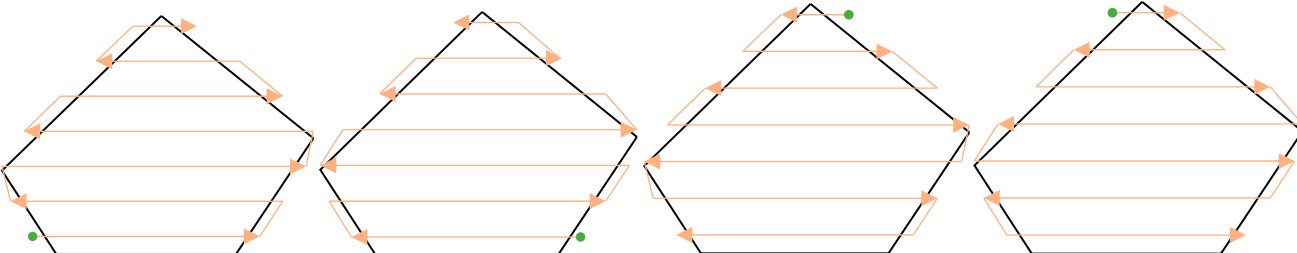

**Figure 6.** Different entry points of the path.

With the inter-regional access sequence firmly established, the judicious arrangement of entry points assumes a paramount role in minimizing path lengths. Therefore, subsequent to the formulation of the optimal inter-regional access sequence, a pragmatic selection is made among these four paths. This selection serves as a strategic step toward ultimately deriving the most optimal overall path configuration.

The pseudocode for generating path points within the region is presented in Algorithm 1. Lines 1–6 encompass the computation of the region's width and determine the optimal flight direction for the UAV. Subsequently, lines 7–25 provide a comprehensive description of how strip-related parameters are calculated along with the determination of endpoints for parallel scanning lines of the UAV. Moving forward, lines 26–30 ascertain both the set of path points and their corresponding lengths at each of the four inlet points. Ultimately, this function yields four distinct paths and their respective lengths within a given region.

---

**Algorithm 1**: *getIntraWay* ($A_i$, $\omega$, $o$)

---

**Input:** Regional parameter $A_i$, sensor parameter $\omega$, $o$
**Output:** Complete collection of waypoints *path*, the path length in the region *dist*

1:     // Get the width of the region and the best flight direction
2:     **For** each $j$ in 1 to $n_i$ **do**
3:       Calculate the distance between the vertex $v_{ij}$ and all edges of the region, and take its maximum value, denoted as $h_j$, and the corresponding edge denoted as $edge_j$
4:     **End for**
5:     $S \leftarrow \min([h_1, h_2, \cdots, h_{n_i}])$
6:     $EDGE \leftarrow$ the edge corresponding to $S$
7:     // Get the scan lines by calculating strip parameters
8:     $\Delta d \leftarrow \omega * (1 - o)$
9:     **If** $S \backslash \Delta d \leq \omega * o$ **then**
10:    $N_b \leftarrow \lfloor S \backslash \Delta d \rfloor$
11:    **Else**
12:    $N_b \leftarrow \lceil S \backslash \Delta d \rceil$
13:    **End if**
14:    **Dor** each $k$ in 1 to $N_b$ **do**
15:      $d_{L_k} \leftarrow \Delta d(k - 1)$
16:      $d_{L_k'} \leftarrow d_{L_k} + \omega$
17:      The length of the strip *len* $\leftarrow$ the maximum value generated by the intersection of the two sides and the middle line of the strip with the region
18:      $L_k \leftarrow$ the coordinates of the two endpoints on one side of the strip are determined by $d_{L_k}$ and *len*
19:      $L_k' \leftarrow$ the coordinates of the two endpoints on the other side of the strip are determined by $d_{L_k'}$ and *len*
20:    **End for**

---

21:     $m_i \leftarrow 2 * N_b$
22:     **For** each $l$ in 1 to $m_i$ **do**
23:        $w_{il} \leftarrow$ the coordinates of the midpoints of the lines connecting corresponding points in $L_k$
          and $L'_k$
24:        $W_i \leftarrow \{W_i, w_{il}\}$
25:     **End for**
26:     // Get the candidate paths based on the four entry points
27:     **For** each $t$ in 1 to 4 **do**
28:        *path(t)* $\leftarrow$ Sort the waypoints in $W_i$ with the $t$-th candidate point as the entry point
29:        *dist(t)* $\leftarrow$ the length of the *path(t)*
30:     **End for**
31:     **Return** *path*, *dist*

In addition to BF coverage, the spiral coverage method can avoid the frequent acceleration and deceleration of the drone during turning, reducing drone jitter. It is an effective method for obtaining stable terrain data. To generate a spiral flight trajectory, for any convex polygonal region, start by selecting adjacent sides among all edges to form perpendicular bisectors, resulting in the same number of intersection points (coordinates may be identical). Take the average of these coordinates, and the resulting point is marked as the reference point for the polygon. Subsequently, connect the vertices of the polygon to the reference point, calculate the lengths of the resulting line segments, and then divide these lengths by the sensor diameter to obtain the segment count. Assuming the UAV enters from the reference point of the polygon, it first flies toward the farthest vertex from that point for a distance equal to the sensor diameter. Then, it flies in the direction of the second-farthest vertex and continues flying in that direction until it exits the polygonal region, as illustrated in Figure 7. The green lines in Figure 7 represent the connections between reference point and vertices, the blue points denote flight waypoints, and the red lines depict the flight path.

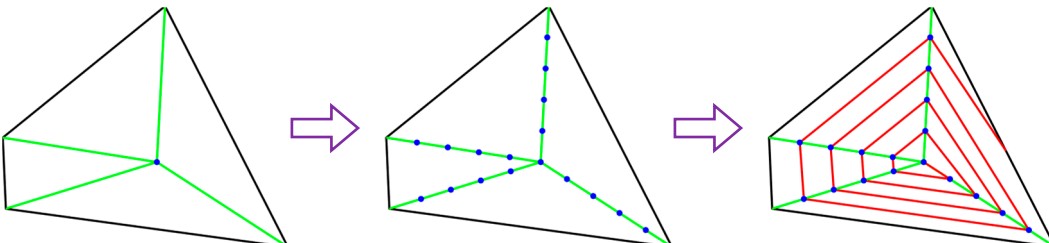

**Figure 7.** Spiral-curve generation method.

However, the UAV's motion capabilities in actual flight are limited, making it challenging to precisely follow sharp corners in simulated routes at turning points. Therefore, smoothing out the sharp corners of the flight path becomes essential to save UAV flight time and reduce jitter during turning.

If there are two points, P0 and P2, around the turning point P1 in the UAV flight path, these three points can be used to form a second-order Bezier curve. This is illustrated in Figure 8a, wherein:

$$\begin{cases} p_{01}(t) = (1-t)P_0 + tP_1 \\ p_{12}(t) = (1-t)P_1 + tP_2 \end{cases} \Rightarrow p_2(t) = (1-t)p_{01} + tp_{12}, \ t \in [0,1]$$

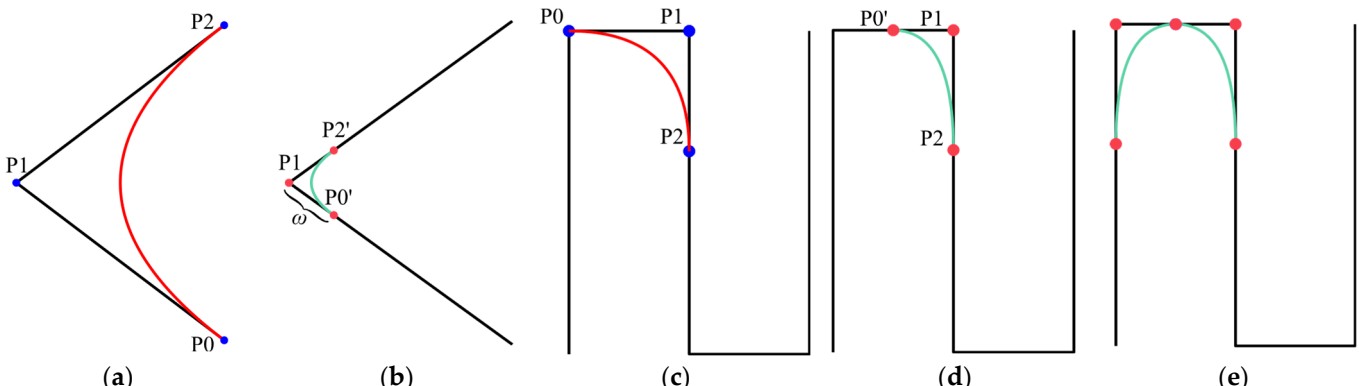

**Figure 8.** Bezier curve and optimized path. (**a**) A second-order Bezier curve. (**b**) Determine control points according to parameter $\omega$. (**c**) A special case. (**d**) The solution. (**e**) Two consecutive paths.

Therefore, the turning point of the UAV flight path is considered as P1. Two points, denoted as P0' and P2', are selected on the two adjacent edges of point P1 at a sensor sampling radius of w. These three points, P0, P1, and P2, act as control points to create a Bezier curve, resulting in a smoothed UAV flight path, as depicted in Figure 8b. This approach helps reduce the jitter during UAV flight. Additionally, the length of the flight path P0→P0'→P2'→P2 is shorter than the straight flight path from P0→P1→P2.

In the method described above, a special case arises when the distance between P0 (or P2) and P1, determined by the sensor sampling radius, w, is not less than the distance between P1 and the adjacent vertices. More precisely, the distance between the two vertices is less than twice the sensor radius, resulting in an incorrect position for point P0 (or P2), as illustrated in Figure 8c. To address this issue, in such cases, the original point P0 (or P2) can be replaced by the midpoint of the adjacent vertices, as shown in Figure 8d. This optimization scheme ensures the correct placement of the point. Two continuous Bezier curves established by the two vertices of this short edge are then depicted in Figure 8e.

When the UAV utilizes a circular sensor for flight with an improved back-and-forth path, the Bezier curve can optimize the flight path, as depicted in Figure 9a, to the improved path shown in Figure 9b, thus optimizing two consecutive turning movements into a single U-turn. When the drone adopts a spiral trajectory, Bezier curves can optimize the flight path shown in Figure 9c to resemble that shown in Figure 9d, smoothing the turning points in the path. Bezier curves are applicable to both of these coverage methods to reduce the length of the flight path and mitigate the jitter phenomenon caused by a too-small turning radius, almost without compromising the coverage effect.

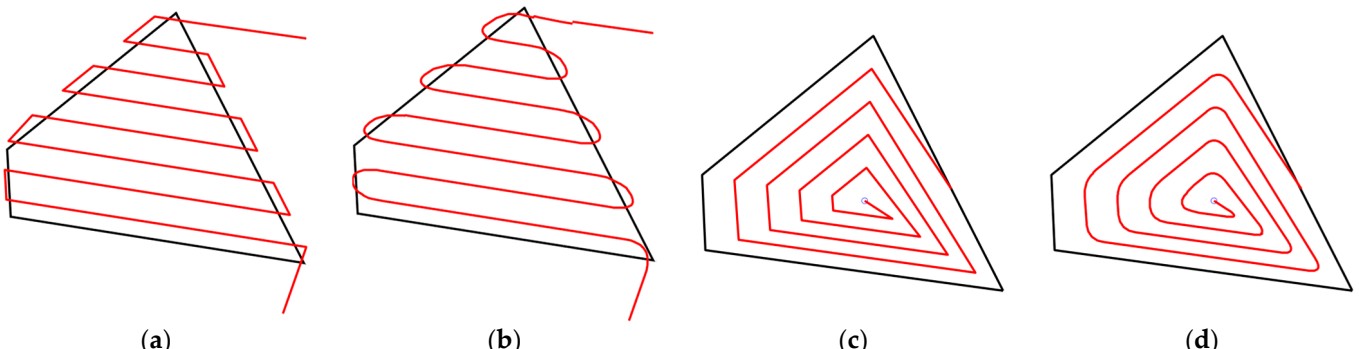

**Figure 9.** Optimization effects of Bezier curves on BF and spiral paths. (**a**) BF path without Bezier. (**b**) BF path with Bezier. (**c**) Spiral path without Bezier. (**d**) Spiral with Bezier.

*3.2. Construction of the Priority-Constrained TSP*

The formulation of the inter-regional access order entails addressing a TSP that accommodates the imposition of priority constraints. In this paper, a meta-heuristic algorithm rooted in the random variable neighborhood descent (RVND) framework [25] is harnessed for this purpose. The algorithmic approach commences by generating an initial solution via the K-nearest neighbors (KNN) heuristic, subsequently adopting the variable neighborhood descent (VND) technique for local search operations. The combination of these strategies effectively yields an approximate optimal solution for the complex problem.

a.　　Two methods of initial solution generation based on PKNN

KNN is a well-established classification algorithm within the realm of machine learning predominantly employed for categorizing samples characterized by similar features. An extension of this, the priority K-nearest neighbors (PKNN) algorithm, draws inspiration from the principles of the KNN algorithm. In the PKNN algorithm, during each iteration, the highest-priority region is chosen from the K-nearest neighboring regions with respect to the current region, serving as the subsequent target for access. The value of K in the PKNN algorithm functions as a limiting factor for the search scope. This algorithm encompasses the capacity to effectively balance priority considerations and path lengths to a reasonable degree.

In this paper, two distinct strategies are advanced for the selection of K-nearest neighbor regions. The following discussion initially outlines the procedural steps intrinsic to the implementation of the first strategy, which is named PKNN-Full, focused on identifying K-neighboring regions across all regions.

Step 1: Introduce a set denoted as $C$ serving as a repository for region numbers that have undergone access alongside a collection $E$ designated to accommodate region numbers that remain unexplored. Simultaneously, establish a collection termed *Order* designed to retain the region numbers, which are systematically arranged according to the computed access order. Given that UAVs are required to initiate coverage operations from a depot, the initialization phase incorporates the inclusion of the depot within the *Order* collection. Proceed to formulate a region matrix denoted as **D** and devise a corresponding priority matrix named **P**. Within **D**, each $i$-th row systematically arranges the sequence numbers of regions in an ascending order, signifying their proximity to region $i$. Simultaneously, **P**, having the same dimensions as **D**, ascribes each individual element to represent the priority attributed to regions occupying corresponding positions in **D**.

Step 2: The last element in the *Order* represents the current region. By consulting matrix **D**, the K-nearest neighboring regions pertaining to the current region are discerned, and their presence in set $C$ is evaluated—this assessment essentially determines whether they have been visited or not. Subsequently, the regions that are yet unvisited are compiled in $E$, effectively emerging as candidate regions. The respective priorities corresponding to these unvisited regions are simultaneously acquired from matrix **P**.

Step 3: Select the region with the highest priority from collection $E$ as the subsequent target for access. Incorporate its sequence number into the *Order* while concurrently removing the region from the set $C$. In instances where the highest-priority regions exhibit a non-unique presence, the region that is proximate to the current region is elected for access.

Step 4: Iterate through the execution of Step 2 and Step 3 until all the regions within set $C$ have been visited, leading to the eventual emptiness of $C$. This iterative process culminates in the attainment of an inter-regional access sequence *Order*.

The ensuing scenarios are anticipated during the implementation of Step 2 and Step 3, each warranting its own resolution strategy, as outlined below:

(1)　If all K-neighboring regions of the current region have been visited, continue to select the subsequent set of K-neighboring regions for the Step 2 operation; if it is found that all such regions have been visited as well, repeat this process until an unvisited region emerges within any group of K-neighboring regions. This unvisited region is

then cataloged within the collection in *E*, subsequently triggering the commencement of Step 3 operations;

(2) In instances where the final set of neighboring regions adjacent to the current region consists of fewer elements than the stipulated value K, it is still treated as a valid set for Step 2 operations;

(3) When all neighboring regions in the current region have been visited, the current region is designated as the last region in the sequence of visits. It is subsequently appended to the *Order* collection, thereby culminating in the aforementioned iterative loop.

The pseudocode for PKNN-Full is presented in Algorithm 2, where lines 1–5 correspond to Step 1, mentioned above. Lines 6–18 constitute a loop body that corresponds to the loop in Steps 2–4, described earlier. Finally, the function outputs the initial solution of the access order.

---

**Algorithm 2**: *PKNN-Full* (*A*, $c_0$, K)

---

**Input:** All regional parameter set *A*, when $A \leftarrow \{A_1 \cdots A_n\}$, the coordinate of depot $c_0$, nearest neighbor parameter K
**Output:** The initial solution of the inter-regional access sequence *Order*

| | |
|---|---|
| 1: | Initialize: Create a set *C* for the unvisited region set where all region numbers are stored, an empty candidate region set *E*, and an empty set *Order* for the access sequence |
| 2: | **For** each *i* in $N_0$ **do** |
| 3: | $\mathbf{D}(i, :) \leftarrow$ region numbers in ascending order of distance from region *i* |
| 4: | $\mathbf{P}(i, :) \leftarrow$ the priority of the region corresponding to the location in $\mathbf{D}$ |
| 5: | **End for** |
| 6: | *Order* $\leftarrow$ |
| 7: | **While** $C \neq \varnothing$ |
| 8: | $cr \leftarrow$ the last element in *Order* |
| 9: | *Item* = 1 |
| 10: | **While** $E \cup C = \varnothing$ |
| 11: | $E \leftarrow \mathbf{D}(cr, (item - 1) \cdot K + 1 : item \cdot K)$ |
| 12: | *Item* $\leftarrow$ *item* + 1 |
| 13: | **End while** |
| 14: | $ar \leftarrow$ the number of the area with the highest priority in *E* according to $\mathbf{P}$ |
| 15: | *Order* $\leftarrow \{Order, ar\}$ |
| 16: | $C \leftarrow C$ set without region *ar* |
| 17: | $E \leftarrow \varnothing$ |
| 18: | **End while** |
| 19: | **Return** *Order* |

---

The second KNN search strategy, referred to as PKNN-Excluded, involves seeking out the K-nearest regions among the unselected regions. In contrast to the aforementioned procedure, the divergence lies solely within the implementation of Step 2. The modified Step 2 operations are delineated as follows:

Step 2: The last element in the set *Order* denotes the current region. Subsequently, matrix **D** is consulted to identify the K-nearest regions that remain unvisited in relation to the current region. These K regions are stored in the collection *E* as candidate regions. Their corresponding priorities are simultaneously extracted from matrix **P**.

The pseudocode of PKNN-Excluded is presented in Algorithm 3, where only lines 9–13 exhibit variations from Algorithm 2 due to the distinct approach employed for neighborhood region selection.

---

**Algorithm 3**: *PKNN-Excluded* (*A*, $c_0$, K)

---

**Input:** All regional parameter set *A*, when $A \leftarrow \{A_1 \cdots A_n\}$, the coordinate of depot $c_0$, nearest neighbor parameter K

**Output:** The initial solution of the inter-regional access sequence *Order*

1:  Initialize: Create a set *C* for the unvisited region set where all region numbers are stored, an empty candidate region set *E*, and an empty set Order for the access sequence

2:  **For** each *i* in $N_0$ **do**

3:  $\quad$ **D**(*i*, :) ← region numbers in ascending order of distance from region *i*

4:  $\quad$ **P**(*i*, :) ← the priority of the region corresponding to the location in **D**

5:  **End for**

6:  *Order* ← {*Order*,0}

7:  **While** $C \neq \varnothing$

8:  $\quad$ *cr* ← the last element in *Order*

9:  $\quad$ **While** $E \cup C = \varnothing$

10: $\quad\quad$ *F* ← **D**(*cr*,:)

11: $\quad\quad$ *F* ← *F* set without region in *Order*

12: $\quad\quad$ *E* ← the first K elements of the set *F*

13: $\quad$ **End while**

14: $\quad$ *ar* ← the number of the area with the highest priority in *E* according to **P**

15: $\quad$ *Order* ← {*Order*, *ar*}

16: $\quad$ *C* ← *C* set without region *ar*

17: $\quad$ *E* ← $\varnothing$

18: **End while**

19: **Return** *Order*

---

The primary objective of the KNN algorithm is to expedite the visitation of regions with higher priority by increasing the value of K. As K varies within the range from 1 to the total number of regions $n - 1$, the search scope progressively extends from nearby regions to encompass all available regions. In situations where the count of searchable regions equals 1, it signifies that, exclusively, the region closest to the current region qualifies for selection as the next target. This approach indeed facilitates ensuring shorter travel distances to a certain extent. However, as the K value gradually increases towards $N - 1$, the scope of searchable regions encompasses the entirety of available regions. In such instances, the next target region is attributed to the region characterized by both the highest priority and proximity to the current location. Ultimately, this methodology guarantees the attainment of optimal priority sequencing. Furthermore, when dealing with regions of identical priority, the principle of minimum distance governs the arrangement of access paths, thereby underlining a comprehensive optimization approach.

Within the context of the two KNN search strategies, PKNN-Full incorporates the regions that have been accessed within the process of identifying K neighbors. This strategy proves advantageous when dealing with a limited number of regions in proximity to the current region. Instead of expending additional distance to uncover regions of higher priority, this strategy endeavors to locate regions in closer proximity, thereby optimizing resource utilization. PKNN-Excluded involves identifying K neighbors by excluding regions that have already been accessed. The advantage of this strategy lies in its immunity to disruption from regions already accessed. This prevents regions of higher priority from being overlooked, ensuring consistent access to the region with higher priority. Both strategies demonstrate flexibility in their application. Therefore, the KNN algorithm incorporating these two search strategies avoids the necessity of rigidly defining a standardized or constant K-value selection logic throughout the research process. Instead, it prioritizes adaptability, dynamically employing the most effective strategy to achieve the highest total score for comprehensive coverage of the target region. This approach is centered on optimizing conditions in response to changing variables. In essence, it strives to generate the finest quality and most gratifying search path, thereby offering invaluable, comprehensive assistance in time-sensitive relief operations functioning under tight time constraints.

b. Local search strategy

Random variable neighborhood descent (RVND) stands as a meta-heuristic algorithm framework proposed by Mladenovi et al. [26]. This framework leverages a diverse range of neighborhood structures, each comprising distinct actions, to facilitate alternating searches, thereby striving for optimal results. Let $t$ denote the number of neighborhood structures and $\{N_1, N_2, \cdots, N_t\}$ denote the set of neighborhood structures. Within the RVND approach, when the present neighborhood structure fails to improve the current optimal solution, the algorithm seamlessly transitions to the subsequent neighborhood to continue the search. The search process concludes once all neighborhoods exhaustively fail to yield improvements to the optimal solution.

In this paper, the approach of the variable neighborhood search (VNS) strategy [20] with a d-relaxed priority constraint is employed to solve the intricacies of access order planning between regions governed by priority constraints. The configuration of the neighborhood structure set, denoted as $N_i$, is underpinned by the d-relaxed priority constraint. The corresponding requirements to satisfy the d-relaxed criteria are outlined below. Importantly, the current operation is performed solely when the constraints corresponding to the pertinent neighborhood structure are successfully satisfied.

(1) Relocated (1)—$N_1$: Reallocate $Order[i]$ to a position succeeding the $j$-th index in the access sequence Order contingent upon the fulfillment of any of the following conditions: ① If $j < i$,

$$p_{j(i-1)}^{min} \geq p_{Order[i]} - d \tag{23}$$

② If $i < j - 1$,

$$p_{(i+1)(j-1)}^{max} \leq p_{Order[i]} + d \tag{24}$$

(2) Relocated (2)—$N_2$: Reallocate $Order[i], Order[i+1]$ to a position succeeding the $j$-th index in the access sequence $Order$ contingent upon the fulfillment of any of the following conditions: ① If $j < i$,

$$p_{j(i-1)}^{min} \geq \max\left(p_{Order[i]}, p_{Order[i+1]}\right) - d \tag{25}$$

② If $j > i + 2$,

$$p_{(i+2)(j-1)}^{max} \leq \min\left(p_{Order[i]}, p_{Order[i+1]}\right) + d \tag{26}$$

(3) Swap (1-1)—$N_3$: Swap $Order[i]$ and $Order[j]$ contingent upon the fulfillment of any of the following conditions: ① If $i < j$,

$$p_{i(j-1)}^{min} \geq p_{Order[j]} - d \text{ and } p_{(i+1)j}^{max} \leq p_{Order[i]} + d \tag{27}$$

② If $j < i$,

$$p_{i(j-1)}^{min} \geq p_{Order[j]} - d \text{ and } p_{(i+1)j}^{max} \leq p_{Order[i]} + d \tag{28}$$

(4) Swap (2-1)—$N_4$: Swap two adjacent regions $Order[i], Order[i+1]$ and another region $Order[j]$ in the access sequence $Order$ contingent upon the fulfillment of any of the following conditions: ① If $j > i + 1$,

$$p_{i(j-1)}^{min} \geq p_{Order[j]} - d \text{ and } p_{(i+2)j}^{max} \leq \min\left(p_{Order[i]}, p_{Order[i+1]}\right) + d \tag{29}$$

② If $j < i$,

$$p_{j(i-1)}^{min} \geq \max\left(p_{Order[i]}, p_{Order[i+1]}\right) - d \text{ and } p_{(j+1)(i-1)}^{max} \leq p_{Order[j]} + d \tag{30}$$

(5) Swap (2-2)—$N_5$: Swap the two adjacent regions $Order[i], Order[i+1]$ and the other two adjacent regions $Order[i], Order[i+1]$ in the access sequence $Order$ contingent upon the fulfillment of the following conditions: If $i + 1 < j$,

$$p_{j(i-1)}^{min} \geq \max\left(p_{Order[i]}, p_{Order[i+1]}\right) - d \text{ and } p_{(i+2)j}^{max} \leq \min\left(p_{Order[i]}, p_{Order[i+1]}\right) + d \tag{31}$$

where $p_{ij}^{min}$ and $p_{ij}^{max}$ respectively represent the minimum and maximum values of priorities from region $i$ to region $j$ in the access sequence *Order*.

c.   Disturbance

When the local search cannot improve the current solution, a perturbation operator comes into play, introducing a random perturbation to guide the current solution away from the local optimality. This study uses a straightforward yet efficacious perturbation strategy encompassing two core operations: Relocated (1) and Swap (1-1). These operations align with the d-relaxed constraints and are executed with distinct selection probabilities of $p$ and $(1 - p)$, respectively.

The primary section of the pseudocode for access order optimization is illustrated in Algorithm 4. Lines 1–2 depict the generation of the initial solution, while the subsequent lines focus on RVND optimization. The function incorporates the five neighborhood structures of d-relaxed constraints as predefined parameters.

---

**Algorithm 4**: *getOrder* (*A*, K)

---

**Input:** All regional parameter set $A$, when $A = \{A_1 \cdots A_n\}$, the coordinate of depot $c_0$, nearest neighbor parameter K
**Output:** The inter-regional access sequence *Order*
1:     *Order* ← *PKNN-Excluded* (*A*, $c_0$, K) or *Order* ← *PKNN-Full* (*A*, $c_0$, K)
2:     *Dist* ← the length of the *Order* is obtained from the center point of the region
3:     *Order'* ← *Order*
4:     **For** each *t* in 1 to 5 **do**
5:      **For** each *i* in 1 to *n* + 1 **do**
6:        **For** each *j* in 1 to n + 1 *do*
7:            *Order'* ← use the neighborhood structure $N_t$ to operate on *Order*
8:            *Dist'* ← the length of the *Order'* is obtained from the center point of the region
9:            **If** *Dist'* < *Dist* **then**
10:             *Order* ← *Order'*
11:            **End if**
12:       **End for**
13:      **End for**
14:     **End for**
15:     **Return** *Order*

---

### 3.3. Selection of Optimal Entry Points and Path Generation

In this paper, drawing inspiration from the principles of the greedy algorithm, a method is devised to determine the entry points for each region based on the inter-regional access order derived in Section 3.2. Specifically, from the four prospective candidate entry points identified in Section 3.1 for each region, the entry point that is closest to the flight point of the previous region is selected as the entry point of the current region. Subsequently, a meticulous arrangement of endpoints for the flying scan lines is orchestrated in alignment with the designated entry points. This arrangement is integral to facilitating the formation of coherent BF scanning path trajectories, effectively constituting the waypoints in the region. This process culminates in the procurement of a comprehensive collection of waypoints spanning multiple regions. These waypoints encompass both the depot and region-based waypoints, organized according to the order of internal access.

The pseudocode for the function that generates the complete path is illustrated in Algorithm 5. This function takes the outputs of Algorithms 1 and 4 as inputs and sequentially connects the intra-regional paths based on regions by selecting appropriate fly-in points. It should be noted that while Algorithm 1 generates intra-regional paths for a single

region, Algorithm 5 requires intra-regional paths for all regions, necessitating running Algorithm 1 separately for each region before executing it.

---

**Algorithm 5**: *getWaypoint* ($c_0$, *Order*, *path*)

---

**Input:** Depot coordinates $c_0$, access sequence *Order*, intra-regional waypoints *path*
**Output:** Complete collection of waypoints *PATH*
1:     $PATH \leftarrow \{PATH; c_0\}$
2:     **For** each $i$ in 1 to $n+1$ **do**
3:       **For** each $j$ in 1 to 4 **do**
4:         $d_j \leftarrow$ the distance between the last coordinate in the *PATH* and the $j$-th entry point in the region *Order (i)*
5:       **End for**
6:       *Path'* $\leftarrow$ the entry point corresponding to the minimum value in $d_1 \cdots d_4$ and the *path* within region *Order (i)* starting from that point
7:     $PATH \leftarrow \{PATH; Path\prime\}$
8:     **End for**
9:     **Return** *PATH*

---

*3.4. Path Evaluation Criteria Based on Priority and Distance*

Given that the paths generated in individual regions adhere to specific length criteria, the evaluation metric in question exclusively assesses the effectiveness of the inter-regional access sequence planning algorithm under the ambit of priority constraints. In this paper, we produce two distinct categories of solutions: distance reference solutions without factoring priority and priority reference solutions without factoring distance. These solutions are generated through specific strategies tailored for each approach. Subsequently, the solution derived from the proposed algorithm—which incorporates both distance and priority considerations—is juxtaposed against these two aforementioned solutions. This comparative process yields the distance score and priority score separately. By summing these scores, the overall score for the current solution is computed. A higher score is indicative of superior optimization outcomes. The subsequent sections outline the methodologies employed for distance scoring and priority scoring.

a.     Distance scoring strategy

As the strategy overlooks the intra-regional path and the inherent regional priority, a simplification is employed, treating each region as a singular particle and regarding the central point's coordinates as the representative location of the region. Furthermore, the task of planning the paths connecting every region to the depot is regarded as a conventional TSP.

The realm of the general TSP problem is well-established and has been extensively studied. Given reasonable constraints on the number and dimensions of regions, numerous algorithms have been developed to uncover the optimal solution. In this paper, a genetic algorithm is employed to obtain the reference solution $Order_{Ref}$ along with its corresponding distance, $Dist_{Ref}$. Comparatively, the optimization algorithm proposed herein yields $Order$ as the solution, accompanied by its respective distance, $Dist$. However, as the algorithm introduced in this paper considers both distance and priority, it necessitates a trade-off, which is manifest as a certain sacrifice in distance length to harmonize the prioritization sequence. As a result, the distance score of the current solution can be expressed as follows:

$$SC_D = \frac{Dist_{Ref}}{Dist} \tag{32}$$

It can be seen that the smaller the distance from *Dist*, the higher the distance score $SC_D$.

b.     Priority scoring strategy

In this paper, we employ the strategy of sequential penalty accumulation for priority scoring. This approach aims to assign a penalty factor, denoted as $Pf$, to each level of

priority. In cases where the priority level is defined as $G = \{1, 2, \cdots, g\}$, the respective penalty factor is designated as $Pf = \{g, g-1, \cdots, 1\}$. This assignment illustrates the relationship between priority and penalty factors, with higher priority levels receiving higher penalty factors in accordance with their importance. For a given access sequence *Order*, let us consider the *i*-th region, whose priority is represented as $p_{Order[i]} = k \in G$, and it corresponds to a specific penalty factor denoted as $Pf_k$. In the context of sequential penalty accumulation, the penalty accumulation process for the *Order* can be defined as follows:

$$T = \sum_{i=1}^{n} i \cdot Pf_k, \; k = p_{Order[i]} \tag{33}$$

Evidently, when the pathway sequence meticulously adheres to the prioritization levels—meaning that regions endowed with higher priorities are addressed foremost—an optimal outcome marked by the minimal sequence penalty accumulation value, $T_{Ref}$, can be achieved. The algorithm expounded within this paper considers both distance and priority assignment, which consequently leads to the emergence of heightened penalty accumulation. This augmented penalty accumulation acts as a counterbalance to the distance aspect when weighed against a solution hewing strictly to priority constraints. Consequently, the formulation for the priority score of the prevailing solution can be articulated as follows:

$$SC_P = \frac{T_{Ref}}{T} \tag{34}$$

c.     Comprehensive scoring strategy

Designating the significance of both spatial distance and priority within the solution, we introduce the distance weight, $w_D$, and the priority weight, $w_P$. In fact, these weights serve to quantify the respective contributions of distance and priority in the solution's evaluation process. Consequently, the comprehensive appraisal of the current solution is encapsulated in the integrated score, which is precisely the weighted mean of the distance score and the priority score:

$$SC = \frac{w_D \cdot SC_D + w_P \cdot SC_P}{w_D + w_P} \tag{35}$$

Within the context of this paper, a balanced consideration between the significance of spatial distance and priority is established by fixing both $w_D$ and $w_P$ at 0.5. This deliberate choice underscores the equivalence of importance attributed to both factors within the solution evaluation framework. Nevertheless, it is important to acknowledge that, in real-world applications, the assignment of the distance weight and the priority weight can be tailored to the precise demands of the scenario at hand. The flexibility to adjust these weights allows for a customized approach that aligns more closely with the specific requirements of the problem under investigation.

## 4. Simulation and Experiment

To commence, the initial step involved generating a collection of 20 convex quadrilaterals, each encircled by an external circle with a radius of 50 m. These quadrilaterals were designated as the target regions necessitating access. Subsequently, a random assignment of priority values, ranging from 1 to 3, was applied to each distinct region. This distribution is visually represented in Figure 10, where priority 1 regions are demarcated by the red zones, priority 2 regions are indicated by the yellow zones, and priority 3 regions are highlighted in green.

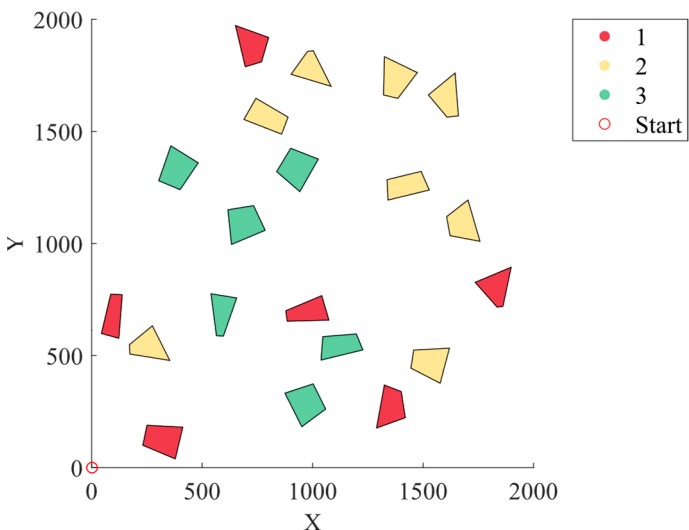

**Figure 10.** Original area.

Within this spatial setup, the point of origin (0, 0) was established as the starting point for the UAV's coverage task. Similarly, the termination point of this coverage task coincided with the same origin. To govern the UAV's sensory reach, a sensor coverage radius spanning 20 m was employed. Moreover, to ensure a judicious sampling process along the UAV's paths, an overlap rate of 0.1 was instituted.

These defined parameters collectively formed the foundation for the simulation. They served as the bedrock upon which the algorithm's efficacy was scrutinized and validated in a simulated environment.

### 4.1. Obtainment of Three Different Initial Solutions

The simulation entailed the execution and analysis of path-planning results for two distinct initial solutions, employing the set of randomly generated regions. Through this simulated process, an intricate evaluation was performed to dissect both the merits and drawbacks of the outcomes. Moreover, a careful examination of the trajectory of scores was undertaken to discern evolving trends.

By subjecting these initial solutions to rigorous simulation, a comprehensive understanding emerged regarding their practical implications. This scrutiny not only dissected the strengths and weaknesses exhibited in the generated paths but also traced the trajectory of scores across the simulation. Such insights contributed significantly to appraising the effectiveness of the path-planning algorithm under scrutiny.

a.     The PKNN-Full Strategy

The evaluation methodology followed the PKNN-Full strategy to derive distinct metrics: the distance score, the priority score, and the cumulative total score for varying K values, as well as the relationship between these scores and the corresponding K values, which were meticulously examined and are visualized in Figure 11.

As illustrated in the figure, a notable trend emerged where the path's length score exhibited a rapid reduction accompanied by a gradual ascent of the priority score towards full realization coinciding with increasing values of K. This pattern can be attributed to the amplified tendency of the path to preferentially select regions of higher priority as target destinations within the process of augmenting K values. This inclination was further substantiated by the positive correlation between higher K values and an increased availability of alternative regions, thereby ensuring earlier selection of regions with elevated priority status.

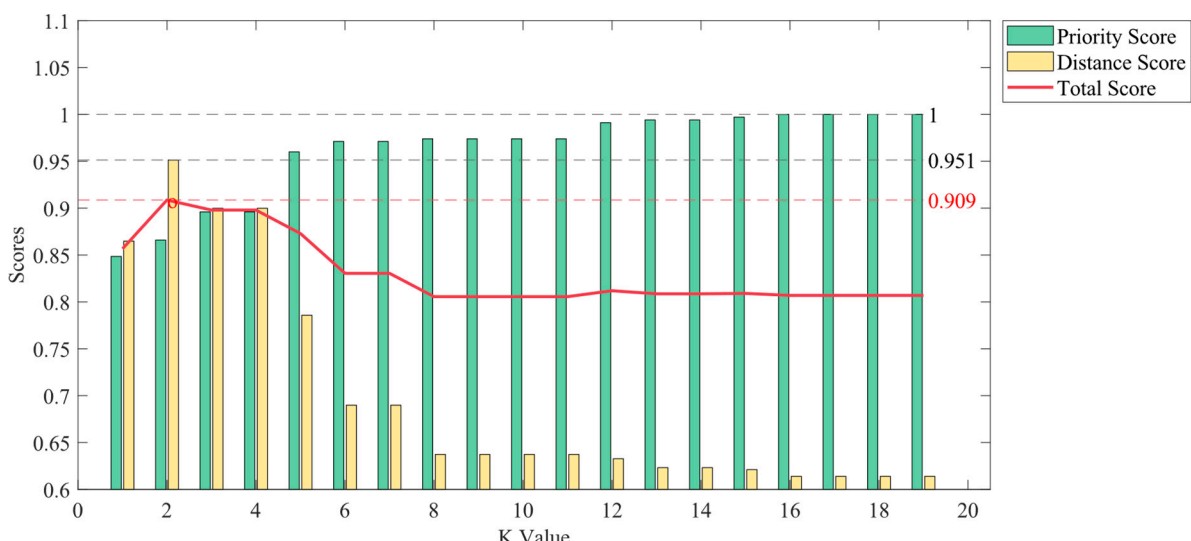

**Figure 11.** PKNN-Full score trend.

Examining the distinctive instances highlighted in the graphical representation, a pivotal observation arises: a zenith in the total score can be observed at K = 2. This specific point signifies the pinnacle performance achievable through the proposed planning methodology. This outcome is indicative of the path length closely approximating the optimal trajectory length while concurrently maintaining a relatively intact priority hierarchy.

Upon closer inspection, for K = 3 and K = 4, a sustained elevation characterizes both scores, manifesting a harmonious equilibrium. This equilibrium proves especially pertinent in scenarios demanding a balanced consideration of both priority and path dynamics, thus adhering to the requisites of balanced solution paradigms.

Upon entering the domain of K = 8, a notable transition can be observed. Here, while the priority score strictly adheres to the higher-priority sequence delineated by distance-based constraints in the selection of a larger set of neighbors, there is a conspicuous decline in the distance score. Consequently, an overall evaluation slightly inferior to the preceding cases ensued.

Beginning at K = 16, a distinctive pattern emerges. The trajectory of region selection becomes stabilized, culminating in the attainment of an optimal priority configuration. Subsequent path planning unfolds meticulously, aligning exactly with the priority order specified by distance-based constraints. This configuration finds particular relevance in environments where priority considerations hold substantial weight.

Guided by the principles underpinning the KNN algorithm, it is evident that, as K approaches $n - 1$, the path progressively attains an optimal priority orientation. To provide a visual portrayal, the comprehensive path corresponding to K = 2 is depicted in Figure 12. The red line in the Figure 12 represents the flight path of the UAV from the depot to the first waypoint, while the remaining flight paths are depicted in black. Various colored polygonal areas represent different priority levels.

b. The PKNN-Excluded Strategy

The evaluation methodology followed the PKNN-Full strategy to derive distinct metrics: the distance score, the priority score, and the cumulative total score for varying K values, as well as the relationship between these scores and the corresponding K values, which were meticulously examined and are visualized in Figure 13.

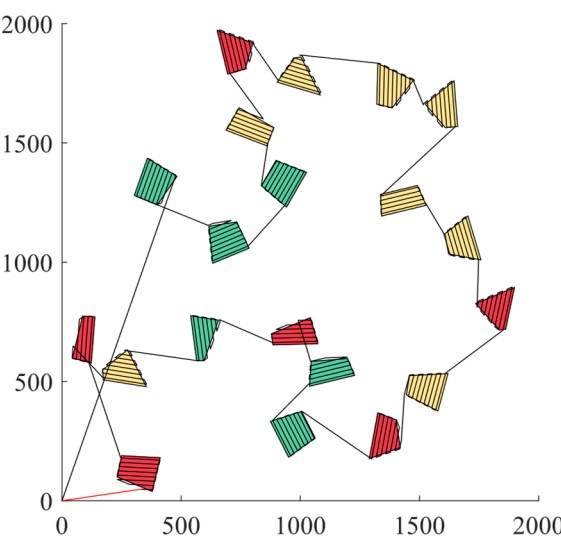

**Figure 12.** Completed path with PKNN-Full.

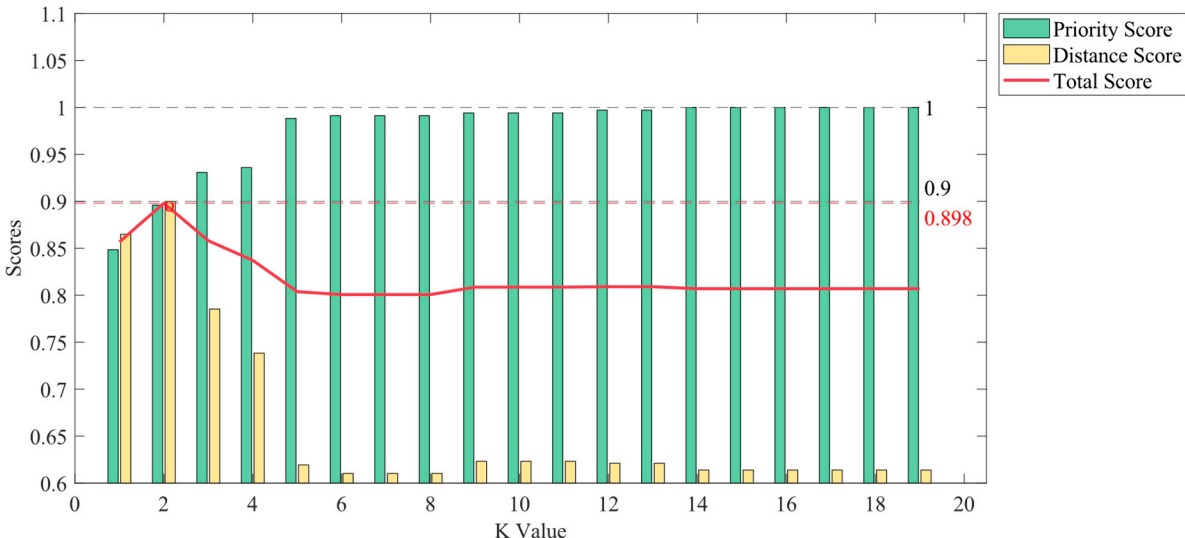

**Figure 13.** PKNN-Excluded score trend.

Figure 13 showcases the outcomes of the KNN algorithm implemented with the adapted search strategy, and, notably, it attained the highest overall score when K = 2. This algorithm variant yielded a path length marginally shorter than that of the previously discussed search methodologies. Nevertheless, it significantly approximated the optimal path length while concurrently elevating the priority, resulting in similar total scores that signified commendable equilibrium. As K = 3, a conspicuous decline in distance becomes apparent. By the onset of K = 5, the priority score experiences a gradual stabilization at an elevated threshold, which also coincides with a deceleration in the descent of the distance score. Echoing the trend, from K = 16 onwards, the trajectory of region selection stabilizes, and its priority reaches its apex configuration. Illustrating this empirically, the comprehensive path for K = 2 is visually presented in Figure 14. The meanings of the colors in Figure 14 are the same as those in Figure 12.

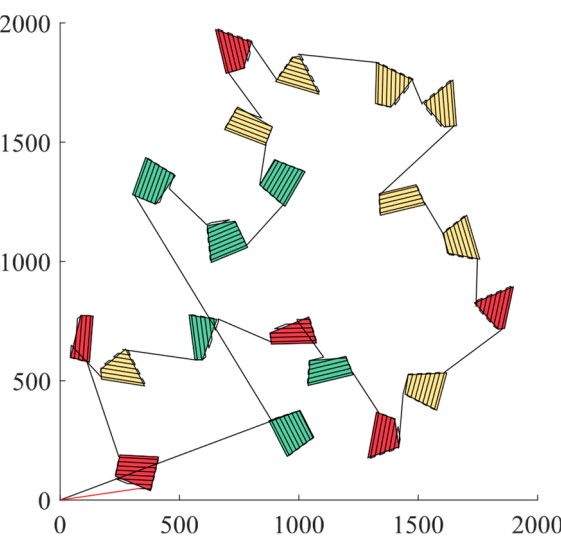

**Figure 14.** Completed path with PKNN-Excluded.

*4.2. Optimizing Initial Solutions by RVND*

Building upon the initial solution, we employed the RVND algorithm for the purpose of optimizing the path. By comparing the outcomes of this optimization process with those of the initial solution, we substantiated the efficacy of the RVND algorithm in curtailing path length while upholding the assured priority score. This comparative analysis served to validate the algorithm's capability in achieving path length reduction without compromising the stipulated priority constraints. The influence of the RVND algorithm on the optimization of the path was examined within the context of the PKNN-Full strategy. Initially, the trend of the total scores was computed across all K values, encompassing the range from 0 to $g - 1$, during d-relaxation. Subsequently, this trend was juxtaposed with the total score progression exhibited by the initial solution, visualized through a line chart, as depicted in Figure 15a. Analogously, the trend of the optimization's total scores for the PKNN-Excluded strategy was also obtained and illustrated in Figure 15b. This comparative analysis served to shed light on the impact of the RVND algorithm on the optimization efficacy of these strategies.

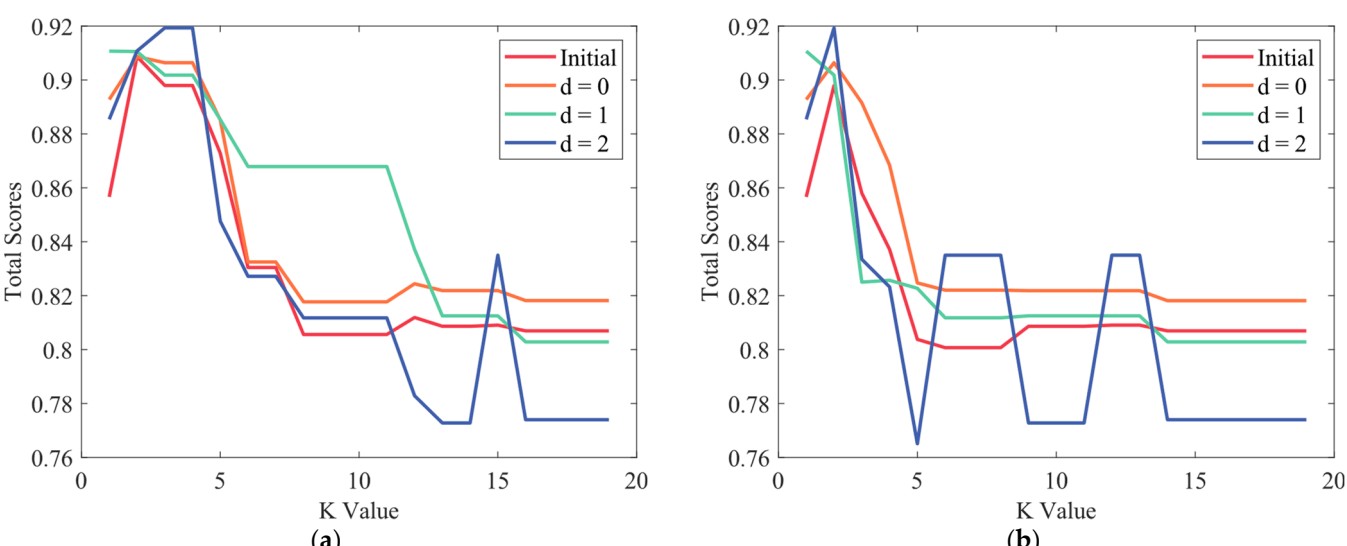

**Figure 15.** Different results of the two strategies: (**a**) d-value comparison using PKNN-Full; (**b**) d-Value comparison using PKNN-Excluded.

The simulation results distinctly illustrate the discernible efficacy of the RVND algorithm in enhancing the initial solutions generated by the PKNN-Full strategy. Particularly noteworthy was its performance in scenarios where $d = 1$, showcasing a pronounced enhancement range across several outcomes and thereby exhibiting conspicuous optimization capabilities. However, in instances where $d = 2$, the magnitude of priority relaxation was substantial, leading to the compromise of priority's significance in favor of a strictly distance-optimized approach. Consequently, given the prevailing distribution of regional priorities, the total score experienced a reduction due to the abrupt depreciation of the priority score.

Similarly, in cases where $d = 0$ and the priority order remained unchanged, the RVND algorithm still displayed discernible path-optimization capabilities. Broadly, the RVND algorithm demonstrably possesses the capacity to optimize path outcomes and is capable of ascertaining relatively optimal pathways tailored to specific requirements. This underscores the algorithm's adaptive capabilities in tailoring solutions in accordance with distinct priorities and demands. The impact of the RVND algorithm on optimizing the initial solution generated by the PKNN-Excluded strategy was notably pronounced. Upon juxtaposition with Figure 15, it becomes evident that this strategy is inherently more inclined to prioritize priority performance. Consequently, its total score exhibits a marginal reduction due to its lower distance score, a characteristic that sets it apart from the Full strategy.

In this context, a notable observation emerges: the disparity between the two algorithms in terms of the maximum value of the initial solution at K = 2 was effectively diminished. Additionally, the value of K derived from the initial solution was optimized towards its maximum value. This phenomenon underscores that an algorithm with modest baseline performance can achieve the same peak outcome after undergoing optimization by a superior algorithm. This manifestation illuminates the robust optimization prowess of the RVND algorithm.

When K = 2 and $d = 1$, the RVND algorithm was used to optimize the two initial solutions, and the complete paths obtained are shown in Figure 16a,b. The meanings of the colors in Figure 16 are the same as those in Figure 12.

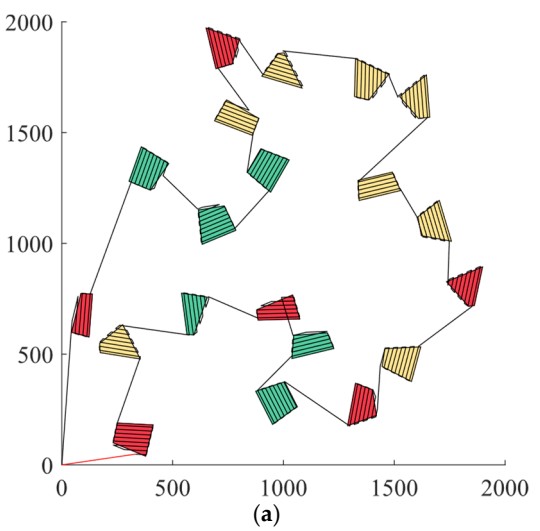
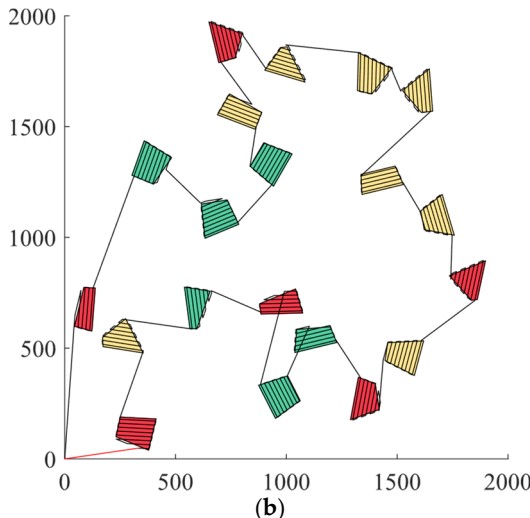

(**a**)　　　　　　　　　　　　　　　　　(**b**)

**Figure 16.** Complete paths obtained with PKNN-Full-RVND and PKNN-Excluded-RVND: (**a**) PKNN-Full-RVND; (**b**) PKNN-Excluded-RVND.

### 4.3. Analysis and Comparison of Optimization Results

a.　Intra-regional path optimization analysis

At the turning points of the intra-regional flight path, replacing the original polyline with Bezier curves and calculating the overall path length as shown in Table 2, through comparative analysis of path length calculations, it can be inferred that Bezier curves can

optimize approximately 5% of the path length without affecting the coverage effectiveness within the region. This indicates a notable energy-saving efficiency. Similarly, applying the research conclusions about Bezier curves mentioned in the introduction, several optimization effects can be observed in the overall UAV path after incorporating Bezier curves:

(1)    Smooth Trajectory: Bezier curves contribute to smoothing the turning angles, reducing the drone's jitter and oscillation during turns, thereby improving flight stability.

(2)    Energy Saving: Bezier curves effectively reduce motion energy consumption in aspects such as path length and motion control, resulting in energy savings for UAV operations.

(3)    Ease of Control: The control method is simple and easy to implement, leading to improved operational efficiency for the UAV.

**Table 2.** Path length optimization rate of Bezier curve.

| Path | PKNN-Full | PKNN-Exclude | PKNN-Full RVND | PKNN-Exclude RVND |
|---|---|---|---|---|
| Original | 29,178.3249 | 29,486.1863 | 28,617.9472 | 28,726.4042 |
| Bezier | 27,553.6353 | 27,862.4824 | 27,003.0889 | 27,121.3444 |
| Opt. (%) | 5.5681 | 5.5067 | 5.6428 | 5.5874 |

b.    Comparison of two optimized coverage methods

Comparing the BF and SP coverage methods, common experimental parameters were set as follows: the minimum circumscribed circle radius of the polygon: 50 m; the sensor scanning radius: 8 m. The number of vertices increased from 4 to 15, with the experiment repeated 20 times for each set of vertices. The average path length for both methods was calculated, and the variation in the average path length is shown in Figure 17. From the figure, it can be observed that, with a small number of polygon edges, the BF path demonstrated better coverage. However, as the number of polygon edges increased, the growth in path length for BF became significantly higher than that for SP. At a vertex count of eight, the two paths were closest, but after reaching nine vertices, the effectiveness of SP surpassed that of BF.

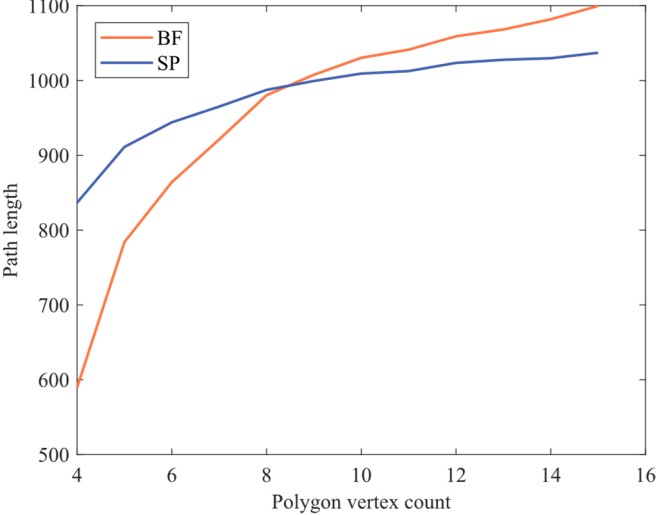

**Figure 17.** Trend: Path length with the number of polygon vertices.

As stated in Section 3.1, both coverage methods can achieve complete coverage within a region, but there will be a certain amount of redundant area. Continuing the analysis of the simulation results, by calculating the coverage area and the polygon area, the redundancy rate of the sensor coverage area for both methods under different vertex conditions was obtained, as shown in Figure 18. It can be observed that the redundancy rate of the BF path

remained stable, while the redundancy rate of the SP path, although substantial when the number of polygon vertices was low, significantly decreased as the vertex count increased. Vertex counts of eight and nine were also critical points for the superiority or inferiority of the two methods in terms of redundancy rate.

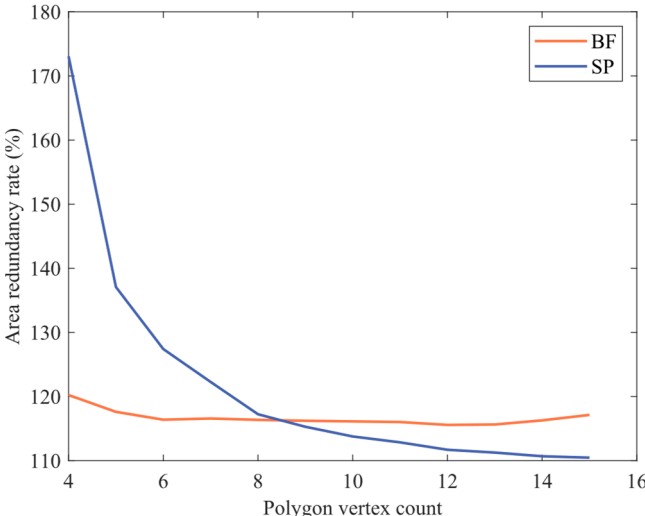

**Figure 18.** Trend: Area redundancy rate with the number of polygon vertices.

From the two aforementioned analytical approaches, it can be observed that if the polygon vertices are evenly distributed around the center of the minimum circumscribed circle of the polygon and when the number of polygon vertices is small, the BF path is likely to have better coverage advantages. It can reduce coverage-area redundancy while obtaining a shorter flight path. However, as the number of polygon vertices increases and the polygon shape gradually becomes smoother, with mostly large internal angles, the UAV is more suitable for using the SP path for coverage flight.

c.    Inter-regional path optimization analysis

Through simulating different numbers of regions and priorities, the optimization performance of the RVND algorithm was analyzed, as presented in Table 3. In this table, min*Dist.* represents the length of the shortest generated path, while max$SC_P$, max$SC_D$, and max$SC$ represent the maximum priority score, the maximum distance score, and the maximum total score, respectively. The term Time denotes the computation time under specific computer performance conditions. Additionally, Opt. signifies the optimization rate of the algorithm towards improving initial solution scores, while Gap indicates the difference in scores between two optimal optimization results.

According to the simulation results recorded in Table 3, when the number of regions and priorities is small, the optimization algorithm exhibits limited effectiveness. This can be attributed to the sufficiency of the generation algorithm for initial solutions in this task, resulting in significantly reduced computation time compared to the optimization algorithm. Hence, when dealing with a small number of regions and priorities, employing two PKNN algorithms can yield higher computational speed. Conversely, when confronted with a large number of priorities, the priority factor plays a more pronounced and crucial role in planning outcomes, thereby highlighting the impact of the optimization algorithm. Considering real-world path considerations, it is recommended to utilize the optimization algorithm for path optimization under conditions involving numerous priorities and regions to achieve superior planning results.

**Table 3.** Summary table of important symbols.

| Params | | | | PKNN-Full | | | | | | PKNN-Excluded | | | | | | Gap |
|---|---|---|---|---|---|---|---|---|---|---|---|---|---|---|---|---|
| No | *n* | *g* | *d* | Min *Dist.* | Max $SC_P$ | Max $SC_D$ | Max $SC$ | Time (*s*) | Opt. (%) | Min *Dist.* | Max $SC_P$ | Max $SC_D$ | Max $SC$ | Time (*s*) | Opt. (%) | F-E (%) |
| 1 | 10 | 2 | - | 7271.9 | 1.000 | 0.995 | 0.937 | 0.003 | 1.17 | 7271.9 | 1.000 | 0.995 | 0.937 | 0.008 | 1.17 | 0 |
| | | | 0 | 7271.9 | 1.000 | 0.995 | 0.937 | 0.028 | | 7271.9 | 1.000 | 0.995 | 0.937 | 0.029 | | |
| | | | 1 | 7232.9 | 0.929 | 1.000 | 0.948 | 0.028 | | 7232.9 | 0.929 | 1.000 | 0.948 | 0.029 | | |
| 2 | 10 | 2 | - | 7244.8 | 1.000 | 0.899 | 0.928 | 0.003 | 3.23 | 7244.8 | 1.000 | 0.899 | 0.863 | 0.003 | 5.79 | +4.93 |
| | | | 0 | 6553.4 | 1.000 | 0.994 | 0.944 | 0.025 | | 6553.4 | 1.000 | 0.994 | 0.910 | 0.028 | | |
| | | | 1 | 6515.4 | 0.938 | 1.000 | 0.958 | 0.029 | | 6515.4 | 0.826 | 1.000 | 0.913 | 0.029 | | |
| 3 | 10 | 2 | - | 7468.4 | 1.000 | 0.927 | 0.918 | 0.003 | 0.87 | 7468.4 | 1.000 | 0.927 | 0.918 | 0.003 | 0.87 | 0 |
| | | | 0 | 7468.4 | 1.000 | 0.927 | 0.918 | 0.028 | | 7468.4 | 1.000 | 0.927 | 0.918 | 0.025 | | |
| | | | 1 | 6920.1 | 0.897 | 1.000 | 0.927 | 0.032 | | 6920.1 | 0.897 | 1.000 | 0.927 | 0.027 | | |
| 4 | 10 | 3 | - | 8323.2 | 1.000 | 0.858 | 0.895 | 0.003 | 6.25 | 8646.7 | 1.000 | 0.826 | 0.895 | 0.003 | 2.23 | +3.93 |
| | | | 0 | 7904.6 | 0.986 | 0.904 | 0.914 | 0.025 | | 8400.7 | 0.973 | 0.850 | 0.899 | 0.261 | | |
| | | | 1 | 7144.2 | 0.973 | 1.000 | 0.951 | 0.029 | | 7144.2 | 0.890 | 1.000 | 0.915 | 0.026 | | |
| | | | 2 | 7144.2 | 0.830 | 1.000 | 0.915 | 0.029 | | 7144.2 | 0.901 | 1.000 | 0.915 | 0.032 | | |
| 5 | 10 | 3 | - | 6948.2 | 1.000 | 0.879 | 0.876 | 0.003 | 8.90 | 6948.2 | 1.000 | 0.879 | 0.878 | 0.003 | 8.66 | 0 |
| | | | 0 | 6735.9 | 0.989 | 0.907 | 0.903 | 0.036 | | 6859.0 | 0.989 | 0.891 | 0.890 | 0.028 | | |
| | | | 1 | 6108.1 | 0.908 | 1.000 | 0.954 | 0.030 | | 6108.1 | 0.908 | 1.000 | 0.954 | 0.050 | | |
| | | | 2 | 6108.1 | 0.908 | 1.000 | 0.954 | 0.033 | | 6108.1 | 0.908 | 1.000 | 0.954 | 0.031 | | |
| 6 | 10 | 3 | - | 7332.5 | 1.000 | 0.945 | 0.859 | 0.003 | 7.92 | 7332.5 | 1.000 | 0.945 | 0.869 | 0.003 | 6.67 | 0 |
| | | | 0 | 7332.5 | 0.967 | 0.945 | 0.873 | 0.024 | | 7332.5 | 0.989 | 0.945 | 0.869 | 0.027 | | |
| | | | 1 | 7332.5 | 0.937 | 0.945 | 0.927 | 0.026 | | 7332.5 | 0.927 | 0.945 | 0.927 | 0.029 | | |
| | | | 2 | 7136.5 | 0.927 | 0.971 | 0.927 | 0.031 | | 7136.5 | 0.918 | 0.971 | 0.903 | 0.029 | | |
| 7 | 20 | 3 | - | 8335.3 | 1.000 | 0.951 | 0.909 | 0.033 | 1.10 | 11,165 | 1.000 | 0.784 | 0.836 | 0.013 | 1.79 | +7.99 |
| | | | 0 | 7930.2 | 1.000 | 0.951 | 0.909 | 0.293 | | 10,827 | 1.000 | 0.808 | 0.851 | 0.280 | | |
| | | | 1 | 7946.9 | 0.966 | 0.998 | 0.911 | 0.338 | | 10,082 | 0.868 | 0.863 | 0.834 | 0.300 | | |
| | | | 2 | 7930.2 | 0.848 | 1.000 | 0.919 | 0.303 | | 9518.7 | 0.819 | 0.919 | 0.842 | 0.303 | | |
| 8 | 20 | 3 | - | 10417.0 | 1.000 | 0.840 | 0.843 | 0.009 | 5.46 | 8812.0 | 1.000 | 0.9 | 0.898 | 0.011 | 2.34 | −3.37 |
| | | | 0 | 9942.3 | 1.000 | 0.880 | 0.875 | 0.275 | | 8582.7 | 1.000 | 0.924 | 0.906 | 0.287 | | |
| | | | 1 | 9856.1 | 0.886 | 0.888 | 0.875 | 0.325 | | 7946.9 | 0.894 | 0.998 | 0.911 | 0.292 | | |
| | | | 2 | 8749.9 | 0.823 | 1.000 | 0.889 | 0.293 | | 8008.8 | 0.848 | 0.990 | 0.919 | 0.314 | | |
| 9 | 20 | 3 | - | 8870.3 | 1.000 | 0.943 | 0.887 | 0.008 | 2.71 | 9043.1 | 1.000 | 0.925 | 0.853 | 0.011 | 6.21 | +0.56 |
| | | | 0 | 8657.8 | 0.997 | 0.967 | 0.897 | 0.284 | | 8772.1 | 0.997 | 0.954 | 0.872 | 0.281 | | |
| | | | 1 | 8369.3 | 0.926 | 1.000 | 0.911 | 0.286 | | 8369.3 | 0.924 | 1.000 | 0.894 | 0.293 | | |
| | | | 2 | 8456.6 | 0.895 | 0.990 | 0.911 | 0.315 | | 8555.6 | 0.895 | 0.978 | 0.906 | 0.306 | | |
| 10 | 20 | 5 | - | 9301.3 | 1.000 | 0.871 | 0.852 | 0.008 | 6.57 | 9301.3 | 1.000 | 0.871 | 0.852 | 0.011 | 6.57 | 0 |
| | | | 0 | 9048.7 | 0.995 | 0.895 | 0.864 | 0.275 | | 9048.7 | 1.000 | 0.895 | 0.864 | 0.277 | | |
| | | | 1 | 8643.9 | 0.941 | 0.937 | 0.890 | 0.291 | | 8643.9 | 0.964 | 0.937 | 0.887 | 0.281 | | |
| | | | 2 | 8175.1 | 0.943 | 0.991 | 0.908 | 0.294 | | 8175.1 | 0.912 | 0.991 | 0.908 | 0.292 | | |
| | | | 3 | 8118.3 | 0.890 | 1.000 | 0.908 | 0.305 | | 8175.1 | 0.916 | 0.991 | 0.908 | 0.298 | | |
| | | | 4 | 8175.1 | 0.849 | 0.991 | 0.908 | 0.288 | | 8175.1 | 0.857 | 0.991 | 0.908 | 0.302 | | |
| 11 | 20 | 5 | - | 9144.6 | 1.000 | 0.821 | 0.786 | 0.007 | 7.00 | 9144.6 | 1.000 | 0.821 | 0.775 | 0.013 | 9.68 | −1.07 |
| | | | 0 | 8940.2 | 0.993 | 0.840 | 0.802 | 0.315 | | 8940.2 | 0.993 | 0.840 | 0.802 | 0.287 | | |
| | | | 1 | 8473.3 | 0.947 | 0.886 | 0.824 | 0.300 | | 8473.3 | 0.933 | 0.886 | 0.850 | 0.295 | | |
| | | | 2 | 8320.6 | 0.912 | 0.903 | 0.841 | 0.307 | | 8320.6 | 0.896 | 0.903 | 0.841 | 0.293 | | |
| | | | 3 | 8285.5 | 0.866 | 0.906 | 0.828 | 0.322 | | 8285.5 | 0.830 | 0.906 | 0.828 | 0.299 | | |
| | | | 4 | 8113.4 | 0.874 | 0.926 | 0.830 | 0.318 | | 8113.4 | 0.874 | 0.926 | 0.826 | 0.318 | | |
| 12 | 20 | 5 | - | 10,569.0 | 1.000 | 0.855 | 0.848 | 0.007 | 2.00 | 11,034 | 1.000 | 0.817 | 0.791 | 0.012 | 8.97 | +0.35 |
| | | | 0 | 10,132.0 | 0.992 | 0.890 | 0.865 | 0.264 | | 10,403 | 1.000 | 0.866 | 0.821 | 0.275 | | |
| | | | 1 | 9938.9 | 0.955 | 0.907 | 0.864 | 0.286 | | 10,266 | 0.955 | 0.878 | 0.851 | 0.290 | | |
| | | | 2 | 10,100.0 | 0.911 | 0.892 | 0.863 | 0.289 | | 10,266 | 0.911 | 0.878 | 0.862 | 0.290 | | |
| | | | 3 | 9790.2 | 0.842 | 0.921 | 0.863 | 0.304 | | 10,278 | 0.867 | 0.877 | 0.858 | 0.298 | | |
| | | | 4 | 9790.2 | 0.801 | 0.921 | 0.846 | 0.293 | | 9941.1 | 0.850 | 0.907 | 0.837 | 0.304 | | |

Although there is no fixed optimal K value due to the influence of regional characteristics, an optimal *d* value can generally be identified from the simulation results: when facing few priorities, a larger *d* value brings about greater distance benefits, thus making it preferable as an optimal *d* value; however, when both region and priority numbers are slightly high simultaneously, the optimal *d* value often appears around the median among all *d* values.

### 4.4. UAV Path-Planning Simulation Platform Based on Unity3D

In order to simulate the flight of a UAV in a realistic environment, we have developed a UAV path-planning simulation system using the Unity3D virtual engine. We import a realistic 3D terrain and drone model and add rigid body components to both the terrain and drone. Additionally, for better realism in simulating the environment, it is necessary to incorporate environmental components, such as wind direction, wind speed, lighting, etc.

Unity3D relies on scripts to implement the operational logic of each object within the virtual environment. Therefore, several scripts need to be added, including region drawing, UAV attitude control, motion trajectory display, sensor data access, etc., along with design of the UI interface and control scripts for UI components. All these scripts work together to ensure the smooth operation of the simulation system. The UI interface of this system is depicted in Figure 19.

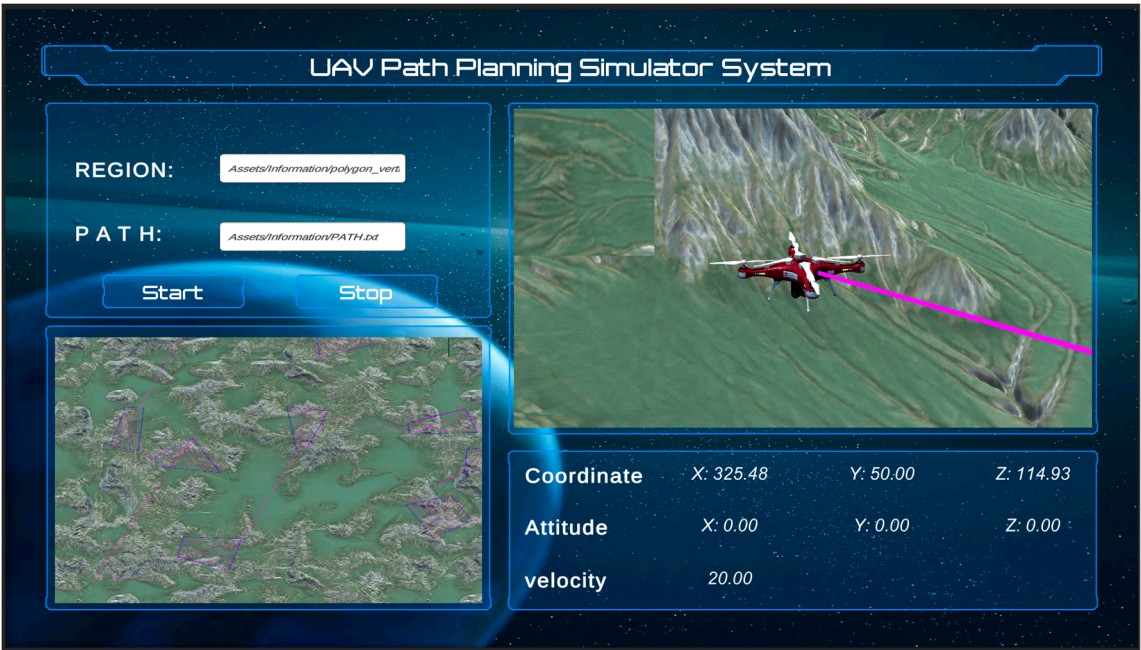

**Figure 19.** UI interface of UAV path-planning simulation system.

The input data for this simulation system consist of path points and area information obtained from MATLAB planning. Upon running the system, it first generates an area range for the terrain that needs coverage; subsequently, the drone flies through this scene based on waypoints while displaying its real-time trajectory. Furthermore, the position and attitude of the drone can also be observed in real time via the UI panel.

### 4.5. Flight-Path Experiment

The experimental section of this study is based on the MATLAB simulation analysis results discussed earlier. Through the use of a UAV for actual coverage flights in a specified area, the effectiveness of adaptive flight trajectories in practical applications was validated, as shown in Figure 20. We selected a spacious environment near the laboratory, as shown in Figure 20a, designating two nearby open areas for overall coverage. These areas were

used to delineate internal regions, and a random location within was chosen as the base position, as illustrated in Figure 20b. In the figure, the region enclosed by the red line is the experimental area, the gray polygon represents the coverage area, and the yellow points indicate the depot of the UAV. The flight path was imported into the UAV remote controller to guide the UAV in flying along the planned path. The resulting flight path in the simulation program is depicted as blue lines in Figure 20c. After the flight experiment, the flight route from the UAV's flight log was exported and is shown as green lines in Figure 20d.

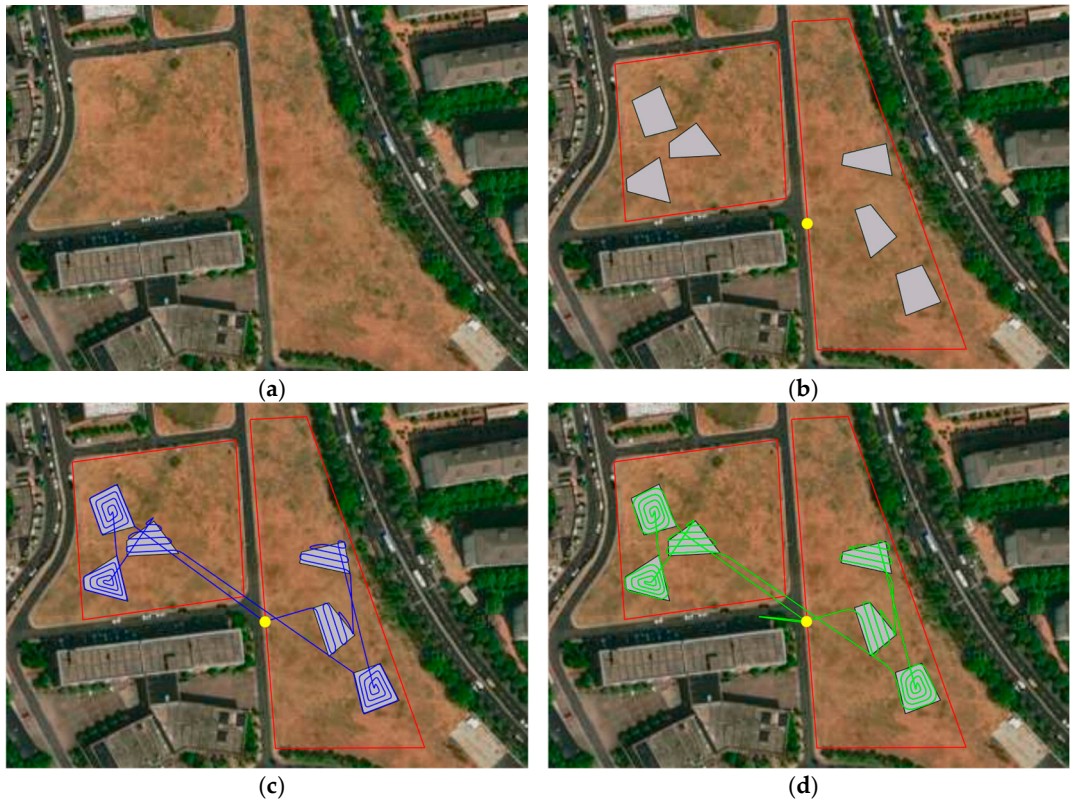

**Figure 20.** Experimental environment and flight experiments. (**a**) The experimental area. (**b**) The drone base, scanning area, and polygonal region. (**c**) The simulated trajectory. (**d**) The experimental trajectory.

Comparative analysis of simulation paths in real environments and paths from actual flight experiments demonstrates that the path-planning algorithm used in this study is well-suited for practical applications in real environments. It effectively achieves UAV path planning within and between areas. The algorithm's performance in real-world applications is thus confirmed. Hence, the flight experiments were successful, as the UAV accurately tracked the theoretical and simulated flight trajectories, validating the practicality of the path-planning method proposed in this study.

## 5. Conclusions

This paper introduces a comprehensive and easily implementable solution to the UAV path-planning problem under priority constraints. We enhance the coverage approach by employing a BF path, ensuring complete coverage within circular sensor sampling ranges and employing Bezier curves to optimize both the round-trip path and the spiral path. Furthermore, we introduce two initial solution generation techniques for priority paths based on the KNN algorithm. These methods are employed to devise the access sequence between regions, incorporating priority considerations. Through comparisons

with planning algorithms lacking priority planning capabilities, our approach demonstrates its ability to intelligently plan paths based on priority orders.

The results of evaluation metrics demonstrate that our proposed method can quickly find high-quality solutions in terms of distance and priority. Furthermore, by optimizing both initial solutions using the RVND algorithm, we enhanced the optimization capabilities of the paths. The simulation results demonstrate the algorithm's strong performance in both distance and priority, indicating its ability to refine solutions from initial states. These outcomes validate the algorithm's effectiveness. Based on our real-world experiments, the algorithm has been demonstrated to exhibit favorable practical prospects in actual application environments. Consequently, the path-planning method presented in this paper holds significant potential for widespread application in the realm of emergency rescue.

Future research will focus on employing intelligent optimization algorithms, such as genetic algorithms, differential evolution, and reinforcement learning, to further enhance the optimization capabilities. It will also work on optimizing CPP path generation strategies to better enable UAVs to cope with external interference. In addition, we plan to explore the use of multi-drone cooperation to simulate more regions and priorities to accomplish complex missions more efficiently.

**Author Contributions:** Conceptualization, L.D. and Y.F.; methodology, L.D.; validation, L.D. and Y.F.; formal analysis, D.Z. and M.G.; investigation, L.D.; resources, D.Z.; data curation, L.D. and Y.F.; writing—original draft preparation, L.D. and Y.F.; writing—review and editing, D.Z. and M.G.; visualization, L.D. and M.G.; supervision, D.Z. and M.G; project administration, D.Z. All authors have read and agreed to the published version of the manuscript.

**Funding:** This research was funded by grants from the National Key Research and Development Program of China (grant no. 2021YFC3090401) and the Open Fund of the Laboratory of Pinghu.

**Data Availability Statement:** Partial data is already included in the charts of the article. The remaining portion of the raw/processed data cannot be shared temporarily as it is part of ongoing research.

**Acknowledgments:** The authors would like to thank the editors and the reviewers for their constructive comments and to thank Chenggong Li and Xingda Zhu for their support during the data collection and analysis.

**Conflicts of Interest:** The authors declare no conflict of interest.

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
