# Peer review of "A Multi-Regional Path-Planning Method for Rescue UAVs with Priority Constraints"

_drones, doi:10.3390/drones7120692_

Round 1
Reviewer 1 Report (Previous Reviewer 1)
Comments and Suggestions for Authors
SUMMARY
This manuscript addresses a multiple Unmanned Aerial Vehicles (UAVs) coverage path planning problem taking into account coverage priorities. For addressing this optimization problem, the Authors propose a new Integer Programming model and a multi-step heuristic solution algorithm
for solving the arising combined Hierarchical Travelling Salesman Problem (HTSP) and Coverage Path Planning (CPP) problems. Results of computational tests are reported and discussed.
RECOMMENDATION
I think this manuscript considers a very relevant optimization problem related to the adoption of UAVs swarm in many engineering applications and offers a new modelling and solution approach that could be worth of interest to a journal like Drones.
In this resubmitted version, the Authors have further refined some passages and addressed my (minor) comments about the decision variables u.
I confirm the recommendation for acceptance expressed in the previous round of revision and I do not have any further comments.
Author Response
Please see the attachment.

Reviewer 2 Report (Previous Reviewer 2)
Comments and Suggestions for Authors
Although Paper is well written, it has a novelty problem in terms of subject matter and does not promise much to the reader. Making single or multi-region cpp is not a very complicated problem. In terms of content, the authors have enriched it as much as they could. However, it is not possible to understand why the authors did not try various methods in a study on simulation. There are many cpp algorithms. https://www.mdpi.com/1424-8220/22/3/1235
Author Response
Please see the attachment.

Reviewer 3 Report (Previous Reviewer 3)
Comments and Suggestions for Authors
This article presents " Multi-Regional Path Planning Method for Rescue UAV with Priority Constraint". The topic is very intriguing due to its applications in the rescue of victims from natural disasters.
The authors need to address some details:
1. Some typos and grammatical errors appear in the text.
2. The authors could provide a bit more detail about the Unity3D simulation. Since the authors mention that realistic conditions were added to the simulation to demonstrate the feasibility of the proposed methodology, they can use this platform to compare the generated ideal route with the actual trajectory executed by the drone.
3. The authors state that there are no relevant works combining HTSP and CPP for comparison. In this regard, they should clarify whether this is because the problem is considered solved through other strategies or if it's simply deemed irrelevant. Additionally, they could discuss the possibility of a comparison using the strategy in which other methods operate and address the problem separately. This would help demonstrate how the proposed methodology enhances specific features or qualities, potentially even creating a performance metric to highlight its strengths.
Comments on the Quality of English LanguageSome typos and grammatical errors appear in the text.
Author Response
"Please see the attachment.

Round 2
Reviewer 2 Report (Previous Reviewer 2)
Comments and Suggestions for Authors
In the study, spiral and scanning structures should be compared. Comparisons should be made in terms of performance. Which method is superior? From where? The paper may be accepted after a sufficient explanation.
Author Response
Please see the attachment.

Reviewer 3 Report (Previous Reviewer 3)
Comments and Suggestions for Authors
Much of my feedback has been addressed.
Author Response
Please see the attachment.

This manuscript is a resubmission of an earlier submission. The following is a list of the peer review reports and author responses from that submission.
Round 1
Reviewer 1 Report
Comments and Suggestions for Authors
Please refer to the attached pdf file.

The quality of English is in general fine.
Reviewer 2 Report
Comments and Suggestions for Authors
The research's primary objective is to optimize the path planning of Unmanned Aerial Vehicles (UAVs) equipped with circular sensors in scenarios where priority constraints dictate which regions to cover first, such as in disaster relief and emergency situations. It achieves this goal through a multi-faceted approach: firstly, by mathematically modeling the problem as a mixed-integer programming challenge, providing a structured framework for solution. Secondly, it introduces innovative improvements to the coverage pattern, particularly catering to circular sensors, ensuring efficient coverage of complex regions. Thirdly, the research proposes two initial solution generation methods based on the K-Nearest Neighbors algorithm to determine the order in which regions are accessed. Fourthly, it employs a meta-heuristic algorithm, RVND, to optimize the access sequence between regions, ultimately enhancing path efficiency and quality. Additionally, a novel evaluation metric, "distance-priority," is introduced to gauge solution effectiveness in terms of both efficiency and priority. Simulation results demonstrate the algorithm's effectiveness in swiftly reaching and covering crucial regions, potentially expediting response times in critical areas. Finally, the research outlines future directions, including the exploration of advanced optimization techniques, improved evaluation criteria, and multi-UAV cooperation for more intricate tasks involving numerous regions and priorities.
Here my comments:
Since the study has an algorithmic structure, benchmarking is indispensable. For this reason, current CPP methods and competitors in this field should be compared clearly in the introduction, which gap in the literature should be closed (is there anyone working on this subject?) and the scientific contribution should be written clearly.
There seems to be a conversion error of L163-170. Line spacing and paragraphs look different. I recommend you review the PDF version.
All track images need to be added in higher resolution.
The work is problem-free in general flow, but the most obvious problem is that it contains a simple CPP problem solution, that is, it is weak and insufficient in terms of innovation and contribution. Alternative and competitor studies must be added and compared. Article need additional experiments For example, the following systems already do more than the recommended method:
https://github.com/Fields2Cover/Fields2Cover
https://github.com/RuslanAgishev/motion_planning
Reviewer 3 Report
Comments and Suggestions for Authors
This article presents "Coverage Path Planning of Rescue UAV for Multi-regional with Priority Constraint". The topic is very intriguing due to its applications in the rescue of victims from natural disasters.
The authors need to address various details and clarify some issues before the work is considered for publication.
1. The contributions of the paper are not clear in the abstract. It would be wise to improve the abstract to highlight the main contributions and proposals of the work.
2. The study assumes that the drone's altitude must remain constant as the environmental perception is achieved using a LIDAR sensor. However, in practice, this is a highly challenging task, especially when the environment is unstructured. The authors should consider that when collecting data using a LIDAR sensor, any change in altitude or tilt of the drone will alter the measurements.
3. The paper presents several contributions to region coverage and the TSP-CPP solution; however, their implementation in real drones is not demonstrated since the test conditions are overly idealistic.
4. The authors should justify the use of a LIDAR sensor for capturing data from regions instead of an RGB or NIR camera.
5. There are several spelling and grammar errors in the text. Additionally, there are many sentences where spaces between words need to be added.

There are several spelling and grammar errors in the text. Additionally, there are many sentences where spaces between words need to be added.
Round 2
Reviewer 1 Report
Comments and Suggestions for Authors
The Authors have addressed all my comments and the overall quality and readability of the manuscript have been improved.
As minor comment, let me point out that, according to me, the definition of the variables u_i is still not fully rigorous: the Authors have added them in the model by declaration (18) and in Table 1, but it would be useful to clearly declare them also in the text before formulas (2) and (3).
Comments on the Quality of English LanguageThe English adopted throughout the text is in general good.
Reviewer 2 Report
Comments and Suggestions for Authors
The study is still weak in terms of scientific contribution, novelty and benchmarking. A simple coverage path planning problem was solved and an experiment was conducted on multiple fields with only the nearest neighbors. For example, why weren't the alternatives in this article tried? https://www.mdpi.com/2504-446X/3/1/4
In addition, authors must provide content such as code, simulation, video, etc. to show that the work is not just about drawings.
Reviewer 3 Report
Comments and Suggestions for Authors
This article presents "Coverage Path Planning of Rescue UAV for Multi-regional with Priority Constraint". The authors have addressed some of the comments, however, there are still details that need to be resolved to prove the relevance of the work.
1. The term "Lidar" is frequently used in the text. However, the accurate representation is "LiDAR," an acronym for either "Light Detection and Ranging" or "Laser Imaging Detection and Ranging."
2. The text contains several spelling and grammatical errors. For instance, the word "aera" is found on line 58.
3. The authors need to specify that when collecting data from environments using a UAV, the appropriate device is a 3D LiDAR, which is notably costly and heavy. Moreover, it's vital for the authors to understand that 3D LiDAR produces a point cloud capturing the terrain's features. While LiDAR is advantageous for reconnaissance and mapping: RGB, NIR, or thermal camera systems are preferable for victim detection during disasters or monitoring.
4. Figure 1 depicts an entirely flat region, making the use of a 3D mapping sensor irrelevant.
5. As highlighted in the introduction, the challenge of covering expansive areas is often addressed with multiple UAV strategies due to their inherent battery constraints. In this regard, the author should discuss a strategy on how their methodology addresses this issue.
6. It would be beneficial to include a comparison against at least two analogous methods to underscore this study's unique advantages.
7. The simulation section should incorporate case studies utilizing standard simulators, like GAZEBO or AirSim.
Comments on the Quality of English LanguageThe text contains several spelling and grammatical errors. For instance, the word "aera" is found on line 58.
